# Interpretable Debiasing of Vectorized Language Representations with Iterative Orthogonalization

**Prince Osei Aboagye[1], Yan Zheng[2], Jack Shunn[1], Chin-Chia Michael Yeh[2], Junpeng Wang[2], Zhongfang Zhuang[2], Huiyuan Chen[2], Liang Wang[2], Wei Zhang[2], Jeff M. Phillips[1]**
[1]University of Utah, [2]Visa Research
[1]{prince,jeffp}@cs.utah.edu, [1]u1161880@utah.edu
[2]{yazheng,miyeh,junpenwa,zhuang,hchen,liawang,wzhan}@visa.com

## ABSTRACT

We propose a new mechanism to augment a word vector embedding representation that offers improved bias removal while retaining the key information—resulting in improved interpretability of the representation. Rather than removing the information associated with a concept that may induce bias, our proposed method identifies two concept subspaces and makes them orthogonal. The resulting representation has these two concepts uncorrelated. Moreover, because they are orthogonal, one can simply apply a rotation on the basis of the representation so that the resulting subspace corresponds with coordinates. This explicit encoding of concepts to coordinates works because they have been made fully orthogonal, which previous approaches do not achieve. Furthermore, we show that this can be extended to multiple subspaces. As a result, one can choose a subset of concepts to be represented transparently and explicitly, while the others are retained in the mixed but extremely expressive format of the representation.

## 1 INTRODUCTION

Vectorized representation of structured data, especially text in Word2Vec (Mikolov et al., 2013), GloVe (Pennington et al., 2014), FastText (Joulin et al., 2016), etc., have become an enormously powerful and useful method for facilitating language learning and understanding. And while for natural language data contextualized embeddings, e.g., ELMO (Peters et al., 2018), BERT (Devlin et al., 2019), RoBERTa (Liu et al., 2019), etc have become the standard for many analysis pipelines, the non-contextualized versions have retained an important purpose for low-resource languages, for their synonym tasks, and their interpretability. In particular, these versions have the intuitive representation that each word is mapped to a vector in a high-dimensional space, and the (cosine) similarity between words in this representation captures how similar the words are in meaning, by how similar are the contexts in which they are commonly used.

Such vectorized representations are common among many other types of structured data, including images (Kiela & Bottou, 2014; Lowe, 2004), nodes in a social network (Grover & Leskovec, 2016; Perozzi et al., 2014), spatial regions of interest (Jenkins et al., 2019), merchants in a financial network (Wang et al., 2021), and many more. In all of these cases, the most effective representations are large, high-dimensional, and trained on a large amount of data. This can be an expensive endeavor, and the goal is often to complete this embedding task once and then use these representations as an intermediate step in many downstream tasks.

In this paper, we consider the goal of adding or adjusting structure in existing embeddings as part of a light-weight representation augmentation. The goal is to complete this without expensive re-training of the embedding but to improve the representation's usefulness, meaningfulness, and interpretability. Within language models, this has most commonly been considered within the context of bias removal (Bolukbasi et al., 2016; Dev & Phillips, 2019). Here commonly, one identifies a linear subspace that encodes some concept (e.g., male-to-female gender) and may modify or remove that subspace when the concept it encodes is not appropriate for a downstream task (e.g., resume ranking). One recently proposed approach of interest called Orthogonal Subspace Correction and

Rectification (OSCaR) (Dev et al., 2021a) identifies two subspaces (e.g., male-female gender and occupations) and performs a continuous deformation of the embedding in the span of those subspaces with the goal of making them, and the concepts they represent, orthogonal.

We build off of this idea for our approach *iterative subspace rectification (ISR)*, but add some subtle but significant modifications and insights:

- We modify how the deformation in the 2-concept subspace takes place. The underlying operation is based on a rotation, and our insight is *how to choose the central point that the data is rotated around*.

- We observe that OSCaR's output representations do not have orthogonal concepts. As such, it can be re-run, *iteratively* – leading to our approach. Using our centered variant, we call this iterative method ISR. It converges so the inherently represented *subspaces are orthogonal*. The uncentered OSCaR does not achieve this convergence.

- Next, we observe that when using this method towards debiasing, ISR *significantly improves the amount of debiasing* compared to all previous methods; e.g., instead of about 50% improvement, ISR attains 95% improvement when measured on the standard WEAT test. When we similarly measure on larger word lists that we generate, the iterative methods we develop are the clear, consistent best performers. With these larger lists, we perform a randomized train-test split experiment (which is rarely performed in this domain), and while the improvements are noisier and less dramatic, our methods are the overall best.

- Moreover, while other debiasing techniques (e.g., Hard Debiasing (Bolukbasi et al., 2016), INLP (Ravfogel et al., 2020)) are based on projections and hence destroy information of the concept for which bias is attenuated (e.g., gender), we can show that ISR *preserves the relevant information*. We evaluate this based on a new measure called Self Word Embedding Association Test (SWEAT).

- Our methods can be extended to *multiple subspace debiasing*, potentially addressing intersectional issues. The resulting representation creates multiple subspaces, all orthogonal.

- Last but not least, the resulting representations are significantly *more interpretable*. After applying this orthogonalization to multiple subspaces, we can perform a basis rotation (that does not change any cosine similarities or Euclidean distances) that results in each of these identified and orthogonalized concepts along a coordinate axis. That is, we maintain the power, flexibility, and compression of a distributed representation, and selected concepts can recover the intuitive and simple coordinate representation of those features. Afterward, these coordinates could simply be ignored on a downstream task if they should not be involved in some aspect of training (e.g., gender for resume sorting) or retained for co-reference resolution. We provide code at https://github.com/poaboagye/ISR-IterativeSubspaceRectification.

**Model of Concepts.** Dating back to the discovery of analogies (e.g., man:woman::king:queen) that encoded the linear structure of word embeddings, an intuitive notion of a concept has been a linear subspace. For instance, the male-female gender subspace as the vector from $v_{man}$ to $v_{woman}$ is consistent with the one from $v_{king}$ to $v_{queen}$. However, this parallel transport does not always generalize.

Instead, in this paper, we follow a slight variant. We posit that a concept is mainly reflected by a set of words that all have high mutual similarity and can simply be represented as the mean point of those sets of words. This would be, for instance, definitionally male words (man, he, his, him, boy, etc) would define one concept and definitionally female ones (woman, she, her, hers, girl, etc) another. Then the male-female gender direction can be defined as the vector between these two means (Dev & Phillips, 2019). Note this explicitly implies *this representation* of gender is binary and does not attempt to (over-)generalize to other non-binary forms of gender; an important task with many challenges (Dev et al., 2021b).

Note that this perspective of how concepts are represented is aligned with the classic WEAT test (Caliskan et al., 2017), which considers the cross-correlation of 4 sets (e.g., male-female vs. math-art). But it diverges from other methods that attempt to model broader concepts such as "nationality" or "occupations" as a single linear subspace but not relying on two polar sets. This perspective is hence slightly less general, but we observe it as more reliable.

## 2  ITERATIVE SUBSPACE RECTIFICATION

Our newly proposed method, Iterative Subspace Rectification (ISR), has three components, each detailed next: centering, rectification (a graded rotation), and iteration.

### 2.1  CENTERING TO CHANGE THE POINT OF ROTATION IN ISR

The key technical challenge in implementing OSCaR (Dev et al., 2021a) is the graded rotation step that rectifies pairs of concepts. The OSCaR method invokes this rotation in the 2-dimensional span of linear concepts $v_1$ (e.g., male-to-female gender) and $v_2$ (e.g., high-to-low power occupations). The point of rotation in this 2-dimensional span is about the origin. We observe that this rotation may not always properly orthogonalize the linear concept vectors. The reason is that if all word vectors defining these concepts are sufficiently far from the origin in roughly the same direction, then it may dilate those word vectors but does not make the resulting linear spans orthogonal. Indeed if one re-learns those representative vectors after this operation, they may increase cosine similarity.

Our proposed solution is to find a central point among the relevant word vectors close to intersecting the learned linear concepts. Then we center the data about this point (shifting all data so that point is the origin) before performing the graded rotation. Like in PCA: we center, perform the analysis/augmentation, and un-center to preserve the original angles with respect to the origin.

The choice of this centering point is simple: as the midpoint of midpoints of concept pairs. In more detail, given two pairs of concepts $A$ and $B$ (e.g., male and female gender) and stereotypical traits or associations $X$ and $Y$ (e.g., unpleasant and pleasant words), we find the mean of each set $\mu(A) = \frac{1}{|A|} \sum_{a \in A} a$ and similarly for $\mu(B), \mu(X), \mu(Y)$. For each pair, we would like to rotate around their midpoints $c_{AB} = (\mu(A) + \mu(B))/2$ and $c_{XY} = (\mu(X) + \mu(Y))/2$. So we choose the center $c$ as the midpoint of those two points as $c = (c_{AB} + c_{XY})/2 = (\mu(A) + \mu(B) + \mu(X) + \mu(Y))/4$. Then after centering, and projecting onto the span of $v_1 = \mu(A) - \mu(B)$ and $v_2 = \mu(X) - \mu(Y)$ those two midpoints $c_{AB}$ and $c_{XY}$ will be close to the origin, especially if the gap $\|c_{AB} - c_{XY}\|$ is small and/or the connecting vector $c_{AB} - c_{XY}$ is nearly orthogonal with $v_1$ and $v_2$.

### 2.2  RECTIFICATION IN ISR

The graded-rotation is the only step that actually augments the data within ISR. It attempts to make orthogonal the identified subspace vectors $v_1$ and $v_2$. Moreover, it applies this operation onto **all** word vector representatives in the data set as a continuous movement. This is essential for two reasons: first, it is (sub-)differentiable, and second, so it generalizes to all other vectorized representations that may carry some of the connotations of a concept but may not be specifically identified as such via a user-supplied word list. For instance, statistically gendered names can represent gender information in these embeddings, but we may not want to specifically assign a gender to the names since people with those names may not associate with the statistically most likely gender.

We leverage the graded-rotation method from OSCaR (Dev et al., 2021a) using their public code. This takes as input two vectors $v_1$ and $v_2$, and all of the vectorized words are projected onto their span and perform a different rotation on each word about the origin. Words close to $v_2$ are rotated to be nearly orthogonal to $v_1$, and words close to $v_1$ are not changed much.

In ISR this is performed after centering the data, and after projecting onto the span of $d$-dimensional vectors $v_1$ and $v_2$. After the rotation, we reconstitute the full-dimensional coordinates of all vectors. Hence this only modifies 2 out of $d$ (e.g., $d = 300$) dimensions in the proper basis, and so the effect on the representation of most word representations is small. The exception is those correlated with the targeted concepts $v_1$ and $v_2$, and as intended, those are updated to become nearly-rectified.

### 2.3  ITERATION IN ISR

The wrapper of ISR is iteration. We find that if we just apply the centering, projection, graded-rotation, un-project, and un-center, the learned subspaces are not completely orthogonal. That is, re-identified $\mu(A), \mu(B), \mu(X)$, and $\mu(Y)$ from the identified word vectors, the vectors $v_1 = \mu(A) - \mu(B)$ and $v_2 = \mu(X) - \mu(Y)$ are not quite orthogonal. However, if we repeat this entire (center-project-rotate-unproject-uncenter) process, then identified vectors quickly approach orthogonality.

## 3 EVALUATION OF DEBIASING AND RECTIFICATION

We first evaluate the effectiveness of ISR in two ways: how well it actually rectifies or orthogonalizes concepts and how well it reduces bias.

Following our models of concepts, all of our methods take as input four word lists: two target word sets $X$ and $Y$ and two sets of attribute words $A$ and $B$. It learns concepts from each pair using their means $\mu(A)$, $\mu(B)$, $\mu(X)$, and $\mu(Y)$ and then the vectors between them $v_1 = \mu(A) - \mu(B)$ and $v_2 = \mu(X) - \mu(Y)$. We found other approaches, such as the normal direction of a linear classifier or the first principal component of the union of a pair, to be less reliable.

**Rectification via Dot Product.** The dot product score measures the level of orthogonality between two linearly-learned concepts. We focus on concepts represented by two sets, $A$ and $B$, and the difference between their two means. Given two such vectors $v_1, v_2 \in \mathbb{R}^d$ we simply compute their Euclidean dot-product as $\langle v_1, v_2 \rangle = v_1^\top v_2 = \|v_1\|\|v_2\| \cos(\theta_{v_1,v_2})$, where $\theta_{v_1,v_2}$ is the angle between the two vectors. If they are orthogonal, the result should be $0$.

**WEAT Score.** The Word Embedding Association Test (WEAT) (Caliskan et al., 2017) was derived from the Implicit Association Test (IAT) from psychology. The goal of WEAT is to measure the level of human-like stereotypical bias associated with words in word embeddings. WEAT uses four sets of words: two target word sets $X$ and $Y$ and two sets of attribute words $A$ and $B$. In short, it computes the average similarity of all pairs, adding those from $X, A$ and $Y, B$, and subtracting otherwise; details in Appendix E. Scores close to 0 indicate no (biased) association, typical values are in $[-2, 2]$.

**Word lists.** Our methods and evaluation methods rely on word lists (and their vectorized forms, unless stated otherwise 300-dimensional GloVe on English Wikipedia (Pennington et al., 2014)). We initially used the standard word list from Caliskan et al. (2017), found in Appendix F. Later we derive and use large word lists from LIWC (Pennebaker et al., 2001), described in Section 3.2.

### 3.1 EVALUATION USING WEAT

As a representative example, we will first explore the relationship between male/female gendered terms and pleasant/unpleasant words. We compare against LP (Dev & Phillips, 2019) (Linear Projection) HD (Bolukbasi et al., 2016) (Hard Debiasing), INLP (Ravfogel et al., 2020) (Iterative Null Space Projection), and OSCaR (Dev et al., 2021a). iOSCaR denotes iteratively running OSCaR and SR as the *non-iterative* subspace rectification with our added centering step. Note that Hard Debiasing includes an equalization step where paired gendered words (e.g., dad-mom) are unprojected and the same distance as they were originally. Such a paired word list concept seems mostly specific to binary-gendered terms, and we simply skip this otherwise.

The WEAT scores are in Table 1, evaluated on the same words used as input to the algorithms. In this case, LP actually increases the WEAT score, and HD, INLP, and OSCaR moderately decrease the scores to about 50% of their previous values. Our method ISR significantly reduces the WEAT score to about 0.03, almost removing all evidence of bias.

Table 1: WEAT Score on Gender Terms vs Pleasant/Unpleasant.

| Orig. | LP | HD | INLP | OSCaR | SR | iOSCaR | ISR |
|-------|-------|-------|-------|-------|-------|--------|-------|
| 0.610 | 0.825 | 0.498 | 0.475 | 0.385 | 0.127 | 0.254  | 0.026 |

We use 10 iterations of subspace rectification; typically, 2-4 is fine. We show the rate of convergence by iteration in Table 2. It also shows the dot product (dotP) scores per iteration. **ISR quickly converges to a very small dot product of** $0$**, so the subspaces are orthogonal**, iOSCaR does not.

We apply similar experiments on many other data set pairs in Table 3; dot products in Appendix G. We observe the ISR normally achieves the smallest (or near-smallest) WEAT score. Occasionally other methods such as INLP (which can remove 35 dimensions from the data set), HD, or LP achieve a competitive amount of bias reduction but typically remove $50 - 90\%$ of the bias. Whereas ISR

Table 2: WEAT Score (WEAT) and Dot Product (dotP) on Gender Terms vs Pleasant/Unpleasant per iteration. ISR converges to orthogonal subspaces (dotP=0), iOSCaR does not.

|  | Before | Iter 1 | Iter 2 | Iter 3 | Iter 4 | Iter 5 | Iter 6 | Iter 7 | Iter 8 | Iter 9 | Iter 10 |
|---|---|---|---|---|---|---|---|---|---|---|---|
| WEAT ISR | 0.610 | 0.127 | 0.010 | 0.018 | 0.024 | 0.025 | 0.026 | 0.026 | 0.026 | 0.026 | 0.026 |
| WEAT iOSCaR | 0.610 | 0.385 | 0.301 | 0.273 | 0.264 | 0.262 | 0.251 | 0.257 | 0.247 | 0.256 | 0.250 |
| dotP ISR | 0.029 | 0.007 | 0.002 | 0.000 | 0.000 | 0.000 | 0.000 | 0.000 | 0.000 | 0.000 | 0.000 |
| dotP iOSCaR | 0.029 | 0.128 | 0.204 | 0.340 | 0.532 | 0.716 | 0.535 | 0.731 | 0.473 | 0.686 | 0.667 |

Table 3: WEAT Score on Pairs of Concepts – using Bespoke Word Lists.

| Concept1 | Concept2 | Orig. | LP | HD | INLP | OSCaR | SR | iOSCaR | ISR |
|---|---|---|---|---|---|---|---|---|---|
| Gen(M/F) | Career/Family | 0.7507 | 0.7713 | 0.2271 | 0.3503 | 0.3343 | 0.3235 | 0.2154 | 0.0114 |
| Gen(M/F) | Math/Art | 0.7302 | 0.6975 | 0.1127 | 0.1262 | 0.5437 | 0.2928 | 0.4435 | 0.0148 |
| Gen(M/F) | Sci/Art | 1.1557 | 0.9068 | 0.1381 | 0.3776 | 0.8642 | 0.4245 | 0.5139 | 0.0140 |
| Name(M/F) | Career/Family | 1.7303 | 0.0421 | 0.0992 | 0.7916 | 0.8950 | 0.6556 | 0.3143 | 0.0186 |
| Name(E/A) | Please/Un | 1.3206 | 0.0800 | 0.0518 | 0.0960 | 0.3043 | 0.7015 | 0.0527 | 0.1678 |
| Flower/Insect | Please/Un | 1.3627 | 0.2395 | 0.1363 | 0.2713 | 0.6348 | 0.3957 | 0.1338 | 0.0254 |
| Music/Weap | Please/Un | 1.4531 | 0.0373 | 0.0942 | 0.0925 | 1.0135 | 0.4728 | 0.2043 | 0.0770 |

commonly removes more than 98% of the bias. Also, note that SR (only one iteration of the centered rectification process) is not nearly as effective as the iterative process in ISR.

## 3.2 EVALUATION USING A TEST / TRAIN SPLIT

The evaluation of debiasing using WEAT with such small and carefully chosen word lists is common. E.g., many papers (Bolukbasi et al., 2016; Dev et al., 2021a) select only the he-she pair to train Gen(M/F). However, a larger goal is to generalize to other words not included in the word lists.

A natural suggestion is to perform cross-validation. That is, split the word lists into two sets at random. Use one set to operate the debiasing mechanism (train) and the other to evaluate on WEAT (test). There are two concerns about this. First, the train-test split approach is predicated on all data points being drawn iid from an underlying distribution, that way, both splits are reflective of that distribution. However, words from natural language are not *iid*; they are in some sense each irreplaceable and unique. Second, the above word lists are rather small, and in halving them, they often become too small to effectively either capture the signal or evaluate the generalization.

We address these concerns (mainly the second) by building larger word lists. We start by pulling categories from LIWC (Pennebaker et al., 2001) that are related to the small bespoke word lists we studied, when possible. We then choose the 100 closest words to the mean of the smaller list. The details and word lists are in Appendix F.

In the following experiments, we perform a 50/50 test/train split on each word list. We perform the debiasing mechanism on the train half and evaluate WEAT on the test half. For each experiment, we repeat it 10 times and report the average value. This captures somewhat how the methods generalize to the concepts at large. However, it does not capture everything as cleanly as the previous (non test/train split) experiment because of the non-*iid* and irreplaceable nature of individual words.

Table 4: WEAT Score on Large Lists and Test/Train Split.

| Concept1 | Concept2 | Orig. | LP | HD | INLP | OSCaR | SR | iOSCaR | ISR |
|---|---|---|---|---|---|---|---|---|---|
| Gen(M/F) | Please/Un | 0.3314 | 0.0331 | 0.2773 | 0.1089 | 0.1030 | 0.1648 | 0.0867 | 0.0872 |
| Gen(M/F) | Career/Family | 0.9079 | 0.6114 | 0.6634 | 0.1838 | 0.5400 | 0.6670 | 0.3658 | 0.3734 |
| Name(M/F) | Please/Un | 1.0427 | 0.0768 | 0.0308 | 0.1900 | 0.1697 | 0.5733 | 0.1882 | 0.2144 |
| Name(M/F) | Career/Family | 1.6617 | 0.2452 | 0.2778 | 0.3971 | 0.0900 | 1.0790 | 0.0364 | 0.4434 |
| Gen(M/F) | Name(M/F) | 1.6796 | 1.3072 | 1.1574 | 0.7505 | 1.5794 | 1.5012 | 0.5789 | 1.0759 |
| Gen(M/F) | Achieve/Anx | 0.8025 | 0.3353 | 0.5057 | 0.1763 | 0.3335 | 0.4771 | 0.2933 | 0.3530 |
| Career/Family | Please/Un | 0.8900 | 0.0416 | 0.1250 | 0.1087 | 0.1217 | 0.3123 | 0.0842 | 0.0346 |
| Career/Family | Achieve/Anx | 1.5344 | 0.0988 | 0.1459 | 0.1543 | 0.3160 | 0.8511 | 0.1914 | 0.3833 |

Table 5: WEAT Score on Large Lists and No Test/Train Split.

| Concept1 | Concept2 | Orig. | LP | HD | INLP | OSCaR | SR | iOSCaR | ISR |
|---|---|---|---|---|---|---|---|---|---|
| Gen(M/F) | Please/Un | 0.3337 | 0.0815 | 0.3368 | 0.1286 | 0.2178 | 0.1089 | 0.1988 | 0.0087 |
| Gen(M/F) | Career/Family | 0.8455 | 0.5793 | 0.4219 | 0.1218 | 0.2296 | 0.4735 | 0.0384 | 0.0116 |
| Name(M/F) | Please/Un | 1.1118 | 0.0311 | 0.0955 | 0.0160 | 0.2694 | 0.5651 | 0.0881 | 0.0377 |
| Name(M/F) | Career/Family | 1.6863 | 0.0061 | 0.0300 | 0.1034 | 0.2117 | 0.9469 | 0.2161 | 0.0046 |
| Gen(M/F) | Name(M/F) | 1.6706 | 1.2107 | 1.2981 | 0.2155 | 1.5656 | 1.2084 | 0.3224 | 0.0066 |
| Gen(M/F) | Achieve/Anx | 0.7477 | 0.2149 | 0.5714 | 0.0320 | 0.0565 | 0.2146 | 0.0866 | 0.0017 |
| Career/Family | Please/Un | 0.9767 | 0.0649 | 0.0031 | 0.1469 | 0.1823 | 0.3576 | 0.1224 | 0.0568 |
| Career/Family | Achieve/Anx | 1.5400 | 0.0696 | 0.1262 | 0.0046 | 0.5245 | 0.6475 | 0.3215 | 0.0386 |

Table 4 shows the results for the test/train split, and Table 5 shows the results for these same large word lists but without the test/train split where the mechanism and evaluation are performed each on the full list. With the test/train split, ISR consistently performs among the best, although there are examples (notably Statistically Gendered Names, Name(M/F)) it does not perform as well. However, in almost all situations where ISR is not the best performing method, another method we propose, iOSCaR (where the non-centered OSCaR is iteratively applied), performs the best. Sometimes some projection-based methods outperform ISR, notably INLP, which iteratively applies projection over 30 times; however, these are also not consistently better than ISR, and especially not iOSCaR. We also observe that, in Table 5, our method ISR does significantly better without the test/train split while other approaches, like iOSCaR, sometimes do about the same. In fact, ISR always has less than a 0.06 WEAT score. We suspect this is because ISR aligns well with this task, and some words are irreplaceable in defining a concept, making test/train split noisy.

## 3.3 EVALUATING BIASES IN PRE-TRAINED LANGUAGE MODELS

Societal biases have also been demonstrated to manifest in large pre-trained contextual language models (May et al., 2019; Kurita et al., 2019; Webster et al., 2020; Guo & Caliskan, 2021; Wolfe & Caliskan, 2021). We evaluate the effectiveness of ISR and iOSCaR at removing such bias on the Sentence Encoder Association Test (SEAT) (May et al., 2019) benchmark. This extends WEAT to contextual representations by constructing semantically neutral template sentences such as "this is a/an [*WORD*]" to create many vectorized representations, of which averages are taken to generate an *effect size* similar to in WEAT. Scores closer to 0 indicate less biased associations.

We consider 3 masked language models (BERT (Devlin et al., 2019), ALBERT (Lan et al., 2020), and RoBERTa (Liu et al., 2019)) and an autoregressive model (GPT-2 (Radford et al., 2019)). Results on ALBERT and GPT-2, and more details of the setup, are deferred to the Appendix B; ALBERT results are similar to BERT and RoBERTa, and GPT-2 exhibits less bias, so the measurements are less meaningful. We present baseline results from Counterfactual Data Augmentation (CDA) (Zmigrod et al., 2019), DROPOUT (Webster et al., 2020), Iterative Nullspace Projection (INLP) (Ravfogel et al., 2020), and SENTENCEDEBIAS (Liang et al., 2020). The last two extend INLP (Ravfogel et al., 2020) and linear projection (Bolukbasi et al., 2016; Dev & Phillips, 2019) to the average of sentences from Wikipedia containing the concept words.

To avoid overfitting concerns, for our methods ISR and iOSCaR, we again use a more extensive word list of size 50. These are chosen among the larger word lists from LIWC (Pennebaker et al., 2001) as the words closest to those in the small sets used in SEAT and WEAT. We then, similar to baselines, vectorize sentences containing those words contained in a Wikipedia dump. We report results for six SEAT tests based on male vs. female gender terms against either Career vs. Family (6), Math vs. Arts (7), or Science vs. Arts (8). The 'b' variants use statistically gendered names instead of definitionally gendered terms.

Table 6, reports the publish effect size of SEAT for the baseline debiasing models from Meade et al. (2022) and our proposed methods iOSCaR and ISR. The original average absolute effect size for BERT and RoBERTa without debiasing is 0.620 and 0.940, respectively, and ISR considerably reduces the effect size to 0.190 and 0.385, respectively. These are the lowest aggregate scores among all of the methods. The next closest scores are typically INLP at 0.204 and 0.823, a technique that removes significant information from the embeddings, unlike ISR.

Finally, as much as ISR is highly effective at mitigating social bias, it is also relatively stable across several tasks evaluated in this paper. This is in contrast to many other debiasing methods, which, as Meade et al. (2022) reported, have a very high variance across different tasks.

Table 6: SEAT test result (effect size) of gender debiased BERT and RoBERTa models. An effect size closer to 0 indicates less (biased) association.

| Model | SEAT-6 | SEAT-6b | SEAT-7 | SEAT-7b | SEAT-8 | SEAT-8b | Avg ($\downarrow$) |
|---|---|---|---|---|---|---|---|
| BERT | 0.931 | 0.090 | $-0.124$ | 0.937 | 0.783 | 0.858 | 0.620 |
| + CDA | 0.846 | 0.186 | $-0.278$ | 1.342 | 0.831 | 0.849 | 0.722 |
| + DROPOUT | 1.136 | 0.317 | 0.138 | 1.179 | 0.879 | 0.939 | 0.765 |
| + INLP | 0.317 | $-0.354$ | $-0.258$ | 0.105 | 0.187 | $-0.004$ | 0.204 |
| + SENTENCEDEBIAS | 0.350 | $-0.298$ | $-0.626$ | 0.458 | 0.413 | 0.462 | 0.434 |
| + iOSCaR (Our approach) | 0.931 | 0.078 | $-1.447$ | $-1.178$ | $-1.21$ | $-1.491$ | 1.056 |
| + ISR (Our approach) | 0.048 | $-0.264$ | $-0.253$ | $-0.035$ | 0.243 | 0.295 | **0.190** |
| RoBERTa | 0.922 | 0.208 | 0.979 | 1.460 | 0.810 | 1.261 | 0.940 |
| + CDA | 0.976 | 0.013 | 0.848 | 1.288 | 0.994 | 1.160 | 0.880 |
| + DROPOUT | 1.134 | 0.209 | 1.161 | 1.482 | 1.136 | 1.321 | 1.074 |
| + INLP | 0.812 | 0.059 | 0.604 | 1.407 | 0.812 | 1.246 | 0.823 |
| + SENTENCEDEBIAS | 0.755 | 0.068 | 0.869 | 1.372 | 0.774 | 1.239 | 0.846 |
| + iOSCaR (Our approach) | 0.894 | 0.268 | 0.574 | 0.648 | 0.504 | 0.729 | 0.603 |
| + ISR (Our approach) | 0.554 | 0.099 | 0.296 | 0.546 | 0.394 | 0.419 | **0.385** |

## 3.4 EVALUATION OF INFORMATION PRESERVED

A critique of the projection-based debiasing mechanisms is that they destroy important information from the vectorized representations. While LP and HD only modify a rank-1 subspace of a very high-dimensional space and, thus, on the whole, do not change the representation that much, INLP may modify a 35-dimensional subspace, which can cause some non-trivial distortions. Moreover, on task-specific challenges (e.g., pronoun resolution involving gender when the male/female gender subspace is removed), significant important information can be lost using the projection-based approaches. In contrast, the orthogonalization-based approaches (OSCaR and the proposed ISR) only skew a rank-2 subspace and so have the potential to retain much more information.

We quantify the task-based information preserved with what we call a Self-WEAT score (or SWEAT score). Given a pair of word lists $A, B$ defining concepts (e.g., Male and Female Terms), we would like to measure how the coherence within each word list $A$ or $B$ compares to the cross-coherence with the other. We can do this by leveraging a random split of each word list and the WEAT score. That is we randomly split $A$ into $A_1$ and $A_2$, and similar $B$ into $B_1$ and $B_2$. Then we compute the WEAT score as WEAT($A_1, B_1, A_2, B_2$). The SWEAT score is the average of this process repeated 10 times. If $A$ and $B$ retain their distinct meaning, this should reflect a similar SWEAT score before and after a debiasing mechanism is applied. If the distinction is destroyed, the SWEAT score will decrease (towards 0) after the debiasing mechanism.

Table 7: SWEAT Score on Large Lists: Measuring Information Preserved.

| Concept1 | Concept2 | Orig. | LP | HD | INLP | OSCaR | SR | iOSCaR | ISR |
|---|---|---|---|---|---|---|---|---|---|
| Gen(M/F) | Please/Un | 1.7674 | 1.2685 | 1.1957 | 0.5528 | 1.5865 | 1.7678 | 0.6424 | 1.7683 |
| Name(M/F) | Please/Un | 1.9041 | 1.0893 | 1.9115 | 0.9475 | 1.8549 | 1.9046 | 1.2711 | 1.9044 |
| Please/Un | Gen(M/F) | 1.8762 | 0.0326 | 1.8862 | 0.7090 | 1.7810 | 1.8759 | 0.8006 | 1.8740 |
| Career/Family | Gen(M/F) | 1.8763 | 0.3530 | 1.8816 | 0.4549 | 1.7720 | 1.8733 | 0.7399 | 1.8527 |
| Achieve/Anx | Gen(M/F) | 1.8677 | 0.5435 | 1.8691 | 0.6893 | 1.7157 | 1.8694 | 0.3939 | 1.8705 |

Table 7 shows the results of several experiments on concept pairs and the effect on the SWEAT score of debiasing. The first concept(Concept1) is the one on which the linear debiasing mechanisms are applied, and the SWEAT score is evaluated, and the second (Concept2) is the concept used in the rotation-based mechanisms. We observe that the pure projection-based mechanisms (LP and INLP) significantly decrease the SWEAT score after debiasing. The hard debiasing mechanism, HD, is

projection based but does not apply projection to the word list used to define the subspace, so it is not surprising that when SWEAT is measured on the same word list, there is typically minimal change to the scores. However, beyond the word lists, the effect would be similar to LP. For instance, note that the Gen(M/F) set corresponds with the original use Bolukbasi et al. (2016), and this overlaps with an *equalize set* of words which we modify their embedding after projection, and while meant to preserve information, it actually decreases the SWEAT score. In contrast, the rotation-based methods, which do not need special restrictions on word lists (especially our method ISR), have almost no decrease in the SWEAT score, hence retaining virtually all of the information pertinent to the two concepts. While OSCaR does not decrease the SWEAT score much, the iterated version iOSCaR exhibits a significant decrease in the SWEAT score, similar to INLP.

**Other Downstream Tasks** To show the effectiveness of our proposed debiasing method, ISR, we also consider other intrinsic and extrinsic tasks. See Appendix C for all the results and details.

## 4 RECTIFICATION OF THREE CONCEPTS

---
**Algorithm 1** 3-ISR$(D, (A, B), (X, Y), (R, S))$
---
1: **for** $k$ iterations **do**
2:     Get concept means: $\mu(A) = \frac{1}{|A|} \sum_{a \in A} a$ and $\mu(B), \mu(X), \mu(Y), \mu(R), \mu(S)$.
3:     Compute center $c$ as
    $c = \frac{\mu(A) + \mu(B) + \mu(X) + \mu(Y) + \mu(R) + \mu(S)}{6}$
4:     Get subspaces: $v_1 = \mu(A) - \mu(B)$, $v_2 = \mu(X) - \mu(Y)$, and $v_3 = \mu(R) - \mu(S)$.
5:     Center all data: $z \leftarrow z - c$ for all $z \in D$.
6:     `Rectify`$(D, v_1, v_2)$
7:     Project: $v_3^{\perp} \leftarrow \text{Span}_{v_1, v_2}(v_3)$.
8:     `Rectify`$(D, v_3^{\perp}, v_3)$.
9:     Uncenter all data: $z \leftarrow z + c$ for all $z \in D$.
10: **return** modified word vectors $D$
---

The proper way to debias word vector embeddings along multiple concepts has long been an important goal. Applying projection-based methods along multiple linearly learned concepts is an option. However, the most effective of these (INLP) removes dozens of dimensions for each concept addressed, so applying it multiple times would start to significantly degrade the information encoded within the embeddings. Another approach Hard Debiasing relies on paired terms (e.g., boy-girl, aunt-uncle) to be explicitly balanced after a projection, but these paired words do not always seem to exist for other concepts.

Using that ISR achieves near-0 dot-products between concepts, we next apply this method iteratively to rectify multiple concepts. As a running example, we consider the issue of how nationality-associated names (from the USA and Mexico) can be associated with gender (and potentially bias that comes with it) as well as with unpleasant sentiments. In this experiment, we will attempt to de-correlate names from these intersectional issues.

**Multiple Subspace Rectification.** As input we take 3 pairs of concepts $A, B$ (e.g., definitionally male/female gendered terms), $R, S$ (e.g., statistically-associated USA/Mexico names), and $X, Y$ (e.g., pleasant / unpleasant terms). As before, for each list we define a mean $\mu(A)$, and for each pair a concept direction $v_1 = \mu(A) - \mu(B)$, $v_2 = \mu(X) - \mu(Y)$, and $v_3 = \mu(R) - \mu(S)$. The goal is to orthogonalize these concepts so that when we recover $v_1$, $v_2$, and $v_3$ from the updated word representations, they are orthogonal. By gradually rotating all data with these words, the premise is that these concepts will de-correlate (and hence de-bias) and retain their internal meaning.

We start by centering at $c = (\mu(A) + \mu(B) + \mu(X) + \mu(Y) + \mu(R) + \mu(S))/6$ the average of all concepts. Then we follow a Gram-Schmidt-style procedure to orthogonalize these concepts. For the pair of concepts with the smallest dot-product (wlog $v_1$ and $v_2$), we run one step of graded rotation. Then we apply this approach on the third concept $v_3$, but with respect to the span of $v_1$ and $v_2$; that is denote $v_3^{\perp}$ as the projection of $v_3$ onto the span of $v_1, v_2$. We then apply a graded-rotation on $v_3$ with respect to $v_3^{\perp}$. Then we uncenter with respect to $c$. This is one iteration, we repeat this entire process for a small number, e.g., 5 iterations.

We outline the procedure in Algorithm 1 which takes as input the word vectors of all words $X$ as well as 3 concept pairs $(A, B)$, $(X, Y)$, $(R, S)$. It leverages the graded-rotation step from OSCaR (Dev et al., 2021a) which we refer to as `Rectify`. This takes in all the word vectors $X$ and two subspace directions $v_1$ and $v_2$. It modifies all points $z \in D$, but only in the span of $v_1$ and $v_2$ so that words aligned with $v_2$ are rotated towards being orthogonal with $v_1$ (within that span) and words aligned with $v_1$ are mostly left as is. We could extend this procedure to more than 3 concept pairs by iteratively applying the `Rectify` method on each of the $j$th subspace $v_j$ with respect to $v_j^\perp = \text{Span}_{v_1, \ldots, v_{j-1}}(v_j)$, the projection onto the span of the previous $j - 1$ directions.

**Evaluation of three subspace rectification.** We evaluate on definitionally gendered male/female terms (GT: $A, B$), pleasant/unpleasant terms (P/U: $X, Y$), and statistically-associated USA/Mexico names (NN: $R, S$), using associated large word lists. GT $v_1$ (gendered terms) and NN $v_2$ (US-A/Mexico) have smallest dot product, so rectify these first within the iteration. Table 8 shows the WEAT score of the ISR mechanism in 5 iterations, measured on the full set; up to iteration 10 is shown in Appendix D.1. All pairwise WEAT scores decrease significantly, with GT vs. NN and GT vs. P/U to about 0.02 and 0.01. The NN vs. P/U has a larger initial value of 1.15 and drops to about 0.14. Also, the pairwise dot products all drop to $< 0.006$. Finally, in Table 8 we show the SWEAT scores for each concept pair. Each concept retains a high self-correlation, preserving their original associations as desired. We performed an additional experiment with three different concepts; see Appendix D.

Table 8: WEAT, dot product, and SWEAT scores for 3-concept debiasing among GT, NN, and P/U.

| | WEAT | | | dot product | | | SWEAT | | |
|---|---|---|---|---|---|---|---|---|---|
| Iteration | GT vs NN | GT vs P/U | NN vs P/U | GT vs NN | GT vs P/U | NN vs P/U | GT | NN | P/U |
| Orig. | 0.1797 | 0.3337 | 1.1506 | 0.0589 | 0.0729 | 0.1721 | 1.7674 | 1.7289 | 1.8762 |
| Iter 1 | 0.1157 | 0.1290 | 0.6195 | 0.0395 | 0.0273 | 0.0598 | 1.7692 | 1.7298 | 1.8768 |
| Iter 2 | 0.0657 | 0.0442 | 0.3146 | 0.0252 | 0.0104 | 0.0204 | 1.7502 | 1.7459 | 1.8648 |
| Iter 3 | 0.0316 | 0.0113 | 0.1974 | 0.0157 | 0.0041 | 0.0070 | 1.7637 | 1.7592 | 1.8715 |
| Iter 4 | 0.0097 | 0.0015 | 0.1564 | 0.0096 | 0.0017 | 0.0024 | 1.7745 | 1.7711 | 1.8761 |
| Iter 5 | 0.0040 | 0.0067 | 0.1423 | 0.0058 | 0.0007 | 0.0008 | 1.7545 | 1.7386 | 1.8603 |

## 5 DISCUSSION

We introduced a new mechanism for augmenting word vector embeddings or any vectorized embedding representations, namely Iterative Subspace Rectification (ISR). It can un-correlate concepts defined by pairs of word lists; this has applications in debiasing and increasing transparency in otherwise opaque distributed representations. While the method is based on a recent method OSCaR (Dev et al., 2021a), it adds some essential extensions that crucially allow the resulting subspaces to be completely orthogonal. In particular, this allows one to post-process the embeddings so the identified concepts can be rotated, an isometric transformation, to be along coordinate axes – allowing a mix of specifically encoded and distributedly encoded aspects of the vector representation.

**Single set concepts.** A major design choice that went into the model of concepts and subspaces, as well as measurement, is that concepts are defined as clusters and subspaces by pairs of clusters; see extended discussion in Appendix A. We also considered an ISR-like method for subspaces defined by single word lists (e.g., occupations). This setting is more general and could potentially be used to rectify concepts that do not have two well-defined polar sets, like occupations or perhaps race, nationality, or ethnicity. We did discover a variant of ISR that empirically converged to a dot-product of 0. This finds the single-set subspace as the top principal component; each of these defines lines $\ell_1$ and $\ell_2$ in $\mathbb{R}^d$. Then to identify a center, it finds the pair of points $p_1 \in \ell_1$ and $p_2 \in \ell_2$ that are as close as possible; this can be solved analytically. The center is chosen as the midpoint of $p_1$ and $p_2$; so $c = (p_1 + p_2)/2$. While this worked fairly well in the sense of dot-product convergence to 0, it was less clear how to evaluate it in terms of bias removal and information retention. Pursuing the generality of this method would be interesting future work.

## 6 ETHICS STATEMENT

Several subtle implementation choices were made for this method to achieve its intended results. For instance, the centering step should occur *before* the projection onto the span of $v_1, v_2$ to perform rectification. Also, in the multiple subspace version of ISR, the iteration loop should wrap around both rectify steps (of $v_1, v_2$ and of $v_3, v_3^\perp$) as opposed to completing one rectification (e.g., iteration of $v_1, v_2$) and then trying to iteratively perform the other ($v_3, v_3^\perp$).

**Limitations.** The main limitation is that the work requires concepts to be easily encoded with a list of words. If the word list is too small, or the relevant words have multiple meanings, then these approaches may prove less effective. An example where stereotypes may occur, but where the community has so far been unable to find suitable word lists to capture concepts include non-binary notions of gender (Dev et al., 2021b). Our work hence focuses on biases occurring in binary representations of gender (male versus female); we remark this not to be exclusionary but to make clear the challenge of addressing non-binary representations of gender (largely its lack of representation in language models) is a limitation of this work. Another related limitation is the way our work addresses nationality. We do so via the most common names at birth from the USA and Mexico. We do not claim this actually encodes the nationality of someone with such a name, but because of the statistical association we draw on to generate these word lists, it serves to encode stereotypes someone with one of these names may face.

**Other considerations.** While under the standard WEAT measurement, our method ISR can virtually eliminate all traces of unwanted associations, however, this complete elimination of measured bias may not transfer to other applications. This is not a new phenomenon (Gonen & Goldberg, 2019; Wang et al., 2020; Dev et al., 2020; Zhao et al., 2019), and for instance, may be the result of bias creeping into the other mechanisms used in the evaluation process. For instance, this may be relevant in downstream tasks where other training data and algorithmic decisions contribute to the overall solution and hence are also subject to bias. Nonetheless, we believe this work has demonstrated significant progress toward eliminating a substantial amount of bias from the core vectorized representation of data.

This work focuses on debiasing of the English language; all evaluation and methods are specified to this context. We hope these ideas generalize to other languages (c.f., Hirasawa & Komachi (2019); Pujari et al. (2019)) as well as vectorized representations of other sorts of data, such as images Kiela & Bottou (2014); Lowe (2004), social networks (Grover & Leskovec, 2016; Perozzi et al., 2014), financial networks (Wang et al., 2021), etc.

Finally, and related to the previous points, we investigate one way of measuring and attenuating bias, focusing on applications in natural language processing. There are, however, other forms of bias, as well as ways to measure and attenuate them.

**On removing bias.** As discussed in the limitation section, this work addresses a limited but highly leveraged form of bias in English language models. Other manifestations and evaluations of bias exist, and it is likely no one methodology or framework can address all aspects. Indeed some may argue that such learned correspondence in representations should not be augmented away. Our method attempts to just orthogonalize the representation of these concepts, still allowing, for instance, a place to have an association with females and pleasant sentiments.

In particular, we focus on concepts captured using polar sets, and this, for instance, may be limiting for groups whose representation does not fit into one of those polar notions and who feel that the unfair treatment is resultant of that representation. Although we have not explicitly attempted to address such a concern, we hope that if there is a set of words that can robustly represent such a group within these word representations, then it can be paired with the complement of that set, and made orthogonal to other concepts, thus removing the unwanted correlation. Identifying and demonstrating this would be important for future work.

Overall, this paper provides a powerful new mechanism for removing unwanted correlations from word vector representations while preserving the existing representation of those concepts. The resulting data representation not only can be shown to dramatically reduce a common bias measurement, but it also increases the interpretation of these representations by allowing multiple identified concepts to occupy coordinate axes.

## 7 REPRODUCIBILITY STATEMENT

All of the Debiasing models run on a CPU. It takes about 4 minutes to run ISR and iOSCaR on a CPU with 10 iterations. Hardware specifications are NVIDIA GeForce GTX Titan XP 12GB, AMD Ryzen 7 1700 eight-core processor, and 62.8GB RAM. All debiasing approaches were completed in under 5 minutes. We used the publicly available codes for existing or baseline debiasing approaches we compared against. We provide links to publicly available codes in references. All the word lists are in Appendix F. We provide code at https://github.com/poaboagye/ISR-IterativeSubspaceRectification.

ACKNOWLEDGMENTS

We thank our support from NSF IIS-1816149, CCF-2115677, and Visa Research.

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

APPENDIX

## A    EXTENDED DISCUSSION

We introduced a new mechanism for augmenting word vector embeddings or any vectorized embedding representations, namely Iterative Subspace Rectification (ISR). It can un-correlate concepts defined by pairs of word lists; this has applications in debiasing and increasing transparency in otherwise opaque distributed representations. While the method is based on a recent method OSCaR (Dev et al., 2021a), it adds some essential extensions that crucially allow the resulting subspaces to be completely orthogonal. In particular, this allows one to post-process the embeddings so the identified concepts can be rotated, an isometric transformation, to be along coordinate axes – allowing a mix of specifically encoded and distributedly encoded aspects of the vector representation.

**Why paired-concept subspaces?**    A major design choice that went into the model of concepts and subspaces, as well as measurement, is that concepts are defined as clusters and subspaces by pairs of clusters. This is more general than subspaces defined by sets of pairs of words (e.g., he-she, man-woman (Bolukbasi et al., 2016)), but not as general as subspaces defined by a single large list of words (e.g., occupations (Dev et al., 2021a)) using their top principal component. This choice was made for two reasons. First, we empirically observed that single word list subspaces were not as stable. For instance, on gendered terms, we could use either approach, and the one that splits the word list into a set of male terms and another set of female terms seemed to be better aligned to the intended direction than just using a single word list. Second, this allowed for a tighter coupling with the evaluation. Previous work sometimes tried to use standard WEAT word lists to evaluate bias removal, but the comparison Male/Female vs. Math/Art may not correlate with the topics used to drive the mechanism Male/Female vs. Pleasant/Unpleasant – so it was an apples to oranges evaluation. Under this setup, we could directly evaluate on the concepts the methods targeted.

## B    EVALUATING BIASES IN PRE-TRAINED LANGUAGE MODELS

The advent of large pre-trained language models has led to remarkable success in several natural language processing (NLP) tasks (Peters et al., 2018; Devlin et al., 2019; Lan et al., 2020; Brown et al., 2020). However, recent works have shown that encoded in pre-trained language models are social biases from the data they were trained on (May et al., 2019; Kurita et al., 2019; Webster et al., 2020; Guo & Caliskan, 2021; Wolfe & Caliskan, 2021). These encoded biases get propagated or amplified by machine learning models in downstream NLP tasks such as machine translation (Stafanovičs et al., 2020; Wang et al., 2022) and sentiment classification (Kiritchenko & Mohammad, 2018) and visual question answering (Goyal et al., 2017; Hudson & Manning, 2019; Hirota et al., 2022). Hence it is imperative to mitigate social biases in pre-trained language models.

Motivated by this, we conduct an experiment to mitigate gender bias in three masked language models (BERT, ALBERT, and RoBERTa) and an autoregressive language model (GPT-2). We evaluated the performance of ISR and iOSCaR against the Sentence Encoder Association Test (SEAT) (May et al., 2019) benchmark.

**Sentence Encoder Association Test (SEAT)**    is a standard intrinsic bias benchmark used to measure the level of bias in pre-trained language models embedding representation. SEAT is an extended version of the Word Embedding Association Test (WEAT) Caliskan et al. (2017) (See Appendix E) to sentence representation, which is particularly useful under pre-trained language models. Similar to WEAT, which measures the stereotypical association between two sets of target concepts and attributes word lists, SEAT substitutes the target and attributes word lists from WEAT into a semantically neutral template such as "this is a/an [*WORD*]" to create target concepts and attributes sentence lists. The vectorized sentence representation is obtained using the average token representation from the last hidden state. After obtaining the sentence vector representation of the two sets of target concepts and attributes, the WEAT test statistic is computed. We report the effect size in the SEAT evaluation. An effect size closer to 0 indicates no (biased) association.

**Baseline Debiasing Models**    Here we describe the four baseline debiasing techniques we compared ISR and iOSCaR against.

- **Counterfactual Data Augmentation (CDA)** (Zmigrod et al., 2019) is a data augmentation technique that re-balances the gendered corpus within the dataset by swapping the male/female attributes to have a more diverse and balanced dataset for the language model pretraining.
- **DROPOUT** (Webster et al., 2020) is a debiasing technique that uses dropout regularization to reduce gender bias by increasing the dropout parameters in the pre-trained language model.
- **Iterative Nullspace Projection (INLP)** (Ravfogel et al., 2020) given your target concept (e.g., male/female gender concept), INLP builds a linear classifier that best separates the target concept and linearly projects all words along the classifier normal.
- **SENTENCEDEBIAS** (Liang et al., 2020) is an extension of linear projection Dev & Phillips (2019) to sentence representation. It starts by identifying the gender direction or subspace and then projects away all the sentence representation from the gender direction or removes the component of the gender subspace for each sentence representation.

**Pretrained Models**   We considered these four pre-trained language models in our gender bias mitigation experiments:

- BERT (Devlin et al., 2019)
- ALBERT (Lan et al., 2020)
- RoBERTa (Liu et al., 2019)
- GPT-2 (Radford et al., 2019)

**Concept Subspace Word List**   We used a more extensive word list of size 50 to determine the concept subspace in the iOSCaR and ISR. We pulled the common categories related to the small SEAT categories from LIWC (Pennebaker et al., 2001); and we then chose the 50 closest words from LIWC to the mean of the smaller list. The details and word lists are in Appendix F.1 (We chose the top 50 words starting from left to right). To contextualize the concept subspace of these other words used in iOSCaR and ISR, we identified their occurrence in sentences within a $2.5\%$ fraction of an English Wikipedia dump. Then we took the average token representation from the last hidden state as the vectorized sentence representation.

**SEAT Test Specifications**   Table 9 provides more details about the SEAT test evaluation.

Table 9: SEAT test specifications (see the original work Caliskan et al. (2017) and Appendix F) for the word list

| Bias type | Test | Demographic-specific words | Stereotype words |
|---|---|---|---|
| Gender | SEAT-6 | Male vs. Female terms | Career vs. Family |
| | SEAT-6B | Male vs. Female names | Career vs. Family |
| | SEAT-7 | Male vs. Female terms | Math vs. Arts |
| | SEAT-7B | Male vs. Female names | Math vs. Arts |
| | SEAT-8 | Male vs. Female terms | Science vs. Arts |
| | SEAT-8B | Male vs. Female names | Science vs. Arts |

**SEAT Results:**   In Table 10, we report the published effect size of SEAT for the baseline debiasing models from Meade et al. (2022) and our proposed debiasing methods, iOSCaR and ISR. The results show that our proposed new debiasing method, ISR, can effectively mitigate gender bias in BERT, ALBERT, RoBERTa, and GPT-2, given the SEAT effect size. The original average absolute effect size for BERT, ALBERT, and RoBERTa without debiasing is 0.620, 0.623, and 0.940, respectively, and ISR considerably reduces the effect size to 0.190, 0.323, and 0.385, respectively. These are the lowest aggregate scores among all of the choices. The next closest scores are typically INLP of 0.204, 0.345, and 0.823, a technique that, unlike ISR, removes significant information from the embeddings. The only other technique that improved upon INLP is the iOSCAR technique we also propose, which reaches 0.603 on the RoBERTa task.

For GPT-2, all debiased models obtain a larger average absolute effect size than the original GPT-2. Again ISR and INLP have the smallest average absolute effect size of 0.138 and 0.119, close to the average absolute effect sizes of 0.113 for GPT-2. As many effect sizes become negative, we suspect measures of these magnitudes are within the typical noise within the evaluation method. This comports with the finding of Guo & Caliskan (2021) that GPT-2 contains the smallest magnitude of overall bias among these contextual models.

Finally, as much as ISR is highly effective at mitigating social bias, it is also relatively stable across several tasks evaluated in this paper. This is excellent since a recent empirical finding from Meade et al. (2022) showed that most debiasing techniques have a very high variance across different tasks. Thus ISR is more stable and able to generalize on various tasks.

We also point out that in this experiment, ISR and iOSCaR are trained on a different (contextualized) word set than is used as the key terms in the evaluation sentences. It is another demonstration of the effectiveness of these methods under a test-train split to show it is not explicitly overfitting.

Table 10: SEAT test result (effect size) of gender debiased BERT, ALBERT, and RoBERTa and GPT-2 models. An effect size closer to 0 indicates no (biased) association.

| Model | SEAT-6 | SEAT-6b | SEAT-7 | SEAT-7b | SEAT-8 | SEAT-8b | Avg ($\downarrow$) |
|---|---|---|---|---|---|---|---|
| BERT | 0.931 | 0.090 | −0.124 | 0.937 | 0.783 | 0.858 | 0.620 |
| + CDA | 0.846 | 0.186 | −0.278 | 1.342 | 0.831 | 0.849 | 0.722 |
| + DROPOUT | 1.136 | 0.317 | 0.138 | 1.179 | 0.879 | 0.939 | 0.765 |
| + INLP | 0.317 | −0.354 | −0.258 | 0.105 | 0.187 | −0.004 | 0.204 |
| + SENTENCEDEBIAS | 0.350 | −0.298 | −0.626 | 0.458 | 0.413 | 0.462 | 0.434 |
| + iOSCaR (Our approach) | 0.931 | 0.078 | −1.447 | −1.178 | −1.210 | −1.491 | 1.056 |
| + ISR (Our approach) | 0.048 | −0.264 | −0.253 | −0.035 | 0.243 | 0.295 | **0.190** |
| ALBERT | 0.637 | 0.151 | 0.487 | 0.956 | 0.683 | 0.823 | 0.623 |
| + CDA | 1.040 | 0.170 | 0.830 | 1.287 | 1.212 | 1.179 | 0.953 |
| + DROPOUT | 0.506 | 0.032 | 0.661 | 0.987 | 1.044 | 0.949 | 0.697 |
| + INLP | 0.574 | −0.068 | −0.186 | 0.566 | 0.161 | 0.518 | 0.345 |
| + SENTENCEDEBIAS | 0.490 | −0.026 | −0.032 | 0.489 | 0.431 | 0.647 | 0.352 |
| + iOSCaR (Our approach) | 0.634 | 0.408 | −0.009 | −0.248 | 0.008 | −0.830 | 0.356 |
| + ISR (Our approach) | 0.511 | −0.067 | 0.280 | 0.268 | 0.483 | 0.329 | **0.323** |
| RoBERTa | 0.922 | 0.208 | 0.979 | 1.460 | 0.810 | 1.261 | 0.940 |
| + CDA | 0.976 | 0.013 | 0.848 | 1.288 | 0.994 | 1.160 | 0.880 |
| + DROPOUT | 1.134 | 0.209 | 1.161 | 1.482 | 1.136 | 1.321 | 1.074 |
| + INLP | 0.812 | 0.059 | 0.604 | 1.407 | 0.812 | 1.246 | 0.823 |
| + SENTENCEDEBIAS | 0.755 | 0.068 | 0.869 | 1.372 | 0.774 | 1.239 | 0.846 |
| + iOSCaR (Our approach) | 0.894 | 0.268 | 0.574 | 0.648 | 0.504 | 0.729 | 0.603 |
| + ISR (Our approach) | 0.554 | 0.099 | 0.296 | 0.546 | 0.394 | 0.419 | **0.385** |
| GPT-2 | 0.138 | 0.003 | −0.023 | 0.002 | −0.224 | −0.287 | **0.113** |
| + CDA | 0.161 | −0.034 | 0.898 | 0.874 | 0.516 | 0.396 | 0.480 |
| + DROPOUT | 0.167 | −0.040 | 0.866 | 0.873 | 0.527 | 0.384 | 0.476 |
| + INLP | 0.106 | −0.029 | −0.033 | −0.015 | −0.236 | −0.295 | 0.119 |
| + SENTENCEDEBIAS | 0.086 | −0.075 | −0.307 | −0.068 | 0.306 | −0.667 | 0.251 |
| + iOSCaR (Our approach) | 0.152 | −0.323 | −0.344 | 0.168 | −0.146 | −0.268 | 0.234 |
| + ISR (Our approach) | 0.079 | −0.133 | 0.014 | 0.187 | −0.086 | 0.329 | 0.138 |

## C  DOWNSTREAM TASK OF DEBIASED WORD EMBEDDINGS

To show the effectiveness of our proposed debiasing method, ISR, beyond intrinsic tasks like WEAT, SEAT, and SWEAT, we also considered other intrinsic tasks, namely Bias-by-Projection (Ding et al., 2022), SemBias Analogy Task (Zhao et al., 2018) and Word Similarity Task, and extrinsic tasks which comprise of POS (part-of-speech) tagging, POS chunking and Named Entity Recognition (NER) (Tjong Kim Sang, 2002).

We considered the small word list to determine the gender and occupation subspace for all the experiments below.

- Gender Words:
    - **Male (10 Words)**: male, man, boy, brother, he, him his, son, sir, masculine
    - **Female (10 Words)**: female, woman, girl, sister, she, her, hers, daughter, madam, feminine
- Occupation Words:
    - **Stereotypically Male (10 Words)**: analyst, scientist, chemist, economist, mathematician, banker, architect, physicist, manager, engineer
    - **Stereotypically Female (10 Words)**: waitress, beautician, maid, housekeeper, receptionist, dancer, hairdresser, choreographer, cook, cashier

### C.1  EXTRINSIC DOWNSTREAM TASK

**NLI Task: Bias Evaluation**    Following the implementation details from Dev et al. (2021a), we perform a natural language inference (NLI) task for bias mitigation. The goal of NLI is to determine if a sentence, i.e., the premise entails, contradicts, or is neutral to another sentence, the hypothesis. Dev et al. (2021a) showed that biased representation could lead to invalid inferences. For instance:



**Premise:** A *doctor* bought a bagel.
**Hypothesis 1:** A *man* bought a bagel.
**Hypothesis 2:** A *woman* bought a bagel.



The question being asked under the inference task above is whether "doctor" implies a male-gendered connotation or a female-gendered one. Both hypothesis sentences are neutral with respect to the premise sentence. However, a language model trained on a biased word embedding predicts entailment for Hypothesis 1 and contradiction for Hypothesis 2 (Parikh et al., 2016). Thus the model says "yes" (entailment), a doctor must be a man, and "no" (contradiction), a doctor can't be a woman.

The aim now is to debiased the word embedding representation and perform an NLI task to measure the level of bias attenuation while maintaining valid gender associations. When debiasing word embeddings, we don't want to alter valid associations, such as between the word pregnant and words like female and mother.

We use the Bias NLI dataset designed by Dev et al. (2020), which consists of $\sim 1.9$ million neutral sentence pairs. They instantiated templates to measure stereotypical inferences with gendered and occupation words. For example:



**Premise:** The *man* age a bagel.
**Hypothesis:** The *accountant* ate a bagel.



A biased or stereotypical inference is measured as a deviation from the neutrality label with metric: Net Neutral (N. Neu), Fraction Neutral (F. Neu), Dev F1, and Test F1. A Higher neutrality score indicates less bias. Net Neutral is the average probability of the neutral label across all sentence pairs, and Fraction Neutral is the fraction of sentence pairs accurately predicted as neutral. A higher N. Neu and F. Neu. score indicates lower bias.

We apply ISR on the first layer of RoBerTa and fine-tune on the Stanford Natural Language Inference (SNLI) dataset (Bowman et al., 2015) and evaluate on the Bias NLI dataset during inference (Dev

et al., 2020). Table 11 shows the N. Neu, F. Neu, Dev F1, and Test F1 scores of the RoBerTa NLI model across various debiasing methods on the before and after debiasing. ISR is slightly outperformed by OSCaR on the neutrality scores (N. Neu and F. Neu) but performs a bit better on the F1 scores on the dev/test sets. Compared to the three other baselines debiasing methods (LP, HD, INLP), ISR achieves higher neutrality and F1 scores except on the Test F1 score, where it is at par with INLP. This shows that ISR significantly reduces bias as OSCaR compared to other debiasing methods.

Table 11: Results on NLI Task for Bias Attenuation. Bias is measured as a deviation from the neutrality label with metric: Net Neutral (N. Neu), Fraction Neutral (F. Neu), Dev F1, and Test F1. A Higher neutrality score indicates lower bias. *: represents results reported from original OSCaR paper Dev et al. (2021a).

| Pretrained Model | Method | N. Neu | F. Neu | Dev F1 | Test F1 |
|---|---|---|---|---|---|
| RoBERTa | Baseline* | 34.9 | 32.1 | 91.2 | 90.5 |
| | LP* | 48.9 | 41.8 | 91.1 | 90.8 |
| | HD* | 45.0 | 35.6 | 91.1 | 90.5 |
| | INLP* | 42.8 | 44.0 | 91.0 | 90.8 |
| | OSCAR* | 56.6 | 58.8 | 91.2 | 90.7 |
| | ISR | 56.2 | 57.8 | 91.5 | 90.8 |

**NLI Task: Information Retention** Similarly, following Dev et al. (2021a), we quantify the level of gender information retention in an extrinsic NLI task. In as much as we want to mitigate bias in vectorized language representations, we do not wish to destroy valid gender associations so that these embeddings retain their utility for other downstream tasks that require robust semantic information.

We follow the experimental setup from Dev et al. (2021a). Likewise, we apply ISR on the first layer of RoBerTa and fine-tune on the Stanford Natural Language Inference (SNLI) dataset (Bowman et al., 2015) and evaluate on the Sentence Inference Retention Test (SIRT) dataset during inference (Dev et al., 2020). Unlike the Bias NLI dataset, which contains neutral sentence pairs, the SIRT dataset contains sentence pairs with ground truth labels, either entailment or contradiction. Each entailment and contradiction dataset contains $\sim 47$ thousand sentence pairs. We measure the quantity of gender information retained using the metrics Net Entail, Fraction Entail, and Net Contradict, Fraction Contradict for the entailment and contradiction datasets, respectively. The Fraction Entail/Contradict score is the accuracy of the model predictions or the fraction of instances for which the model predicts the entailment/contradiction class. Net Entail/Contradict measures the average probability of the entailment/contradiction label across all sentence pairs.

Table 12: Results on the Gendered Information Preserved under an NLI Task. The degree of information retention is measured with the SIRT (sentence inference retention test) metric: N. Ent, F. Ent, N. Con, and F. Con. Higher scores indicate more gendered information is retained. *: represents results reported from original OSCaR paper Dev et al. (2021a).

| Pretrained Model | Method | N. Ent | F. Ent | N. Con | F. Con |
|---|---|---|---|---|---|
| RoBERTa | Baseline* | 94.9 | 98.4 | 97.4 | 97.7 |
| | LP* | 95.9 | 99.7 | 98.9 | 99.4 |
| | HD* | 95.1 | 98.6 | 98.7 | 99.3 |
| | INLP* | 92.8 | 97.1 | 95.4 | 96.4 |
| | OSCaR* | 95.1 | 99.0 | 99.4 | 99.7 |
| | ISR | 97.4 | 100.0 | 99.5 | 99.8 |

As expected, the original RoBERTa embedding without debiasing (Baseline) does well on the SIRT test with net entail and fraction entail at 94.9 and 98.4, and net contradict and fractional contradict at 97.4 and 97.7, an indication of significantly preserving the gendered information correctly. LP,

OSCaR, and HD perform similarly to the Baseline except for INLP, where we see a drop in all four scores (N. Ent, F. Ent, N. Con, and F. Con).

ISR is the best-performing method compared to the other debiasing technique (LP, HD, INLP, and OSCaR) and the Baseline on the SIRT test. We see a significant improvement with net entail and fraction entail at 97.4 and 100.0 and net contradict and fractional contradict at 99.5 and 99.8. Thus we see an almost perfect gendered information preservation with ISR. This is, therefore, in the synergy of our goal to improve bias removal while retaining the key information.

**CoNLL2003 Shared Task**  To investigate the debiasing impacts of our proposed debiasing method, ISR, on its ability to still retain a good extrinsic downstream utility and performance in standard natural language processing (NLP) task, we considered the CoNLL2003 shared task (Tjong Kim Sang, 2002). Under the CoNLL2003 shared task, we use POS (part-of-speech) tagging, POS chunking and Named Entity Recognition (NER) as the three evaluation tasks following Manzini et al. (2019). Each task is evaluated in two ways: 1) Embedding Matrix Replacement — We first train the specific task model on the biased word embedding, and at test time, we compute the evaluation metric difference between using the biased embeddings and the debiased embeddings, and 2) Model Retraining — Here we train two separate models for a given evaluation task. One on the biased word embeddings and the other on the debiased word embeddings, and at test time, we compute the difference in the performance of these two models. A positive value for the Embedding Matrix Replacement and Model Retraining experiments means the task performs better than the original biased embedding.

The results are shown in Table 13. Under the Embedding Matrix Replacement experiment, ISR and INLP outperform all the other debiasing techniques across all the evaluation tasks and evaluation metrics. Thus they achieve no decrease in performance except precision in POS Tagging for INLP and Recall in POS Chunking for ISR.

Also, in Model Retraining, ISR is the second best performing debiasing technique after INLP. ISR improves performance for F1 and Recall in POS Tagging and POS Chunking, and Recall in NER.

Overall, ISR shows stable and comparable performance across the three tasks. This signifies that sematic downstream utility is preserved under ISR.

Table 13: Downstream tasks of POS Tagging, POS Chunking, and Named Entity Recognition. A positive value means the task performs better than original biased embedding and $\Delta$ represents the change before and after debiasing.

| | Embedding Matrix Replacement | | | | | | | | |
|---|---|---|---|---|---|---|---|---|---|
| | POS Tagging | | | POS Chunking | | | Named Entity Recognition | | |
| | $\Delta$ F1 | $\Delta$ Precision | $\Delta$ Recall | $\Delta$ F1 | $\Delta$ Precision | $\Delta$ Recall | $\Delta$ F1 | $\Delta$ Precision | $\Delta$ Recall |
| LP | 0.0009 | -0.0004 | -0.0025 | -0.0007 | -0.0011 | -0.0016 | -0.0004 | -0.0002 | -0.0013 |
| HD | -0.0009 | 0.0000 | -0.0029 | -0.0009 | -0.0010 | -0.0022 | -0.0005 | 0.0000 | -0.0015 |
| INLP | 0.0001 | -0.0005 | 0.0006 | 0.0003 | 0.0004 | 0.0006 | 0.0001 | 0.0000 | 0.0003 |
| ISR | 0.0000 | 0.0000 | 0.0002 | 0.0000 | 0.0003 | -0.0002 | 0.0001 | 0.0000 | 0.0003 |
| | Model Retraining | | | | | | | | |
| | POS Tagging | | | POS Chunking | | | Named Entity Recognition | | |
| | $\Delta$ F1 | $\Delta$ Precision | $\Delta$ Recall | $\Delta$ F1 | $\Delta$ Precision | $\Delta$ Recall | $\Delta$ F1 | $\Delta$ Precision | $\Delta$ Recall |
| LP | 0.0027 | -0.0052 | 0.0111 | 0.0002 | 0.0033 | -0.0020 | -0.0012 | -0.0056 | 0.0006 |
| HD | -0.0052 | -0.0127 | -0.0086 | 0.0000 | -0.0102 | 0.0075 | -0.0007 | -0.0057 | 0.0024 |
| INLP | 0.0030 | 0.0020 | 0.0079 | 0.0046 | -0.0359 | 0.0439 | -0.0014 | -0.0123 | 0.0062 |
| ISR | 0.0003 | -0.0043 | 0.0033 | 0.0017 | -0.0102 | 0.0142 | -0.0004 | -0.0049 | 0.0032 |

### C.2 Intrinsic Downstream Tasks

To confirm the effectiveness of our method beyond intrinsic measures or metrics like the WEAT, SEAT, and SWEAT, we also run some other intrinsic evaluation methods. Namely, Bias-by-Projection (Ding et al., 2022), SemBias Analogy Task (Zhao et al., 2018) and Word Similarity Task. We also compared the performance of our proposed debiasing method to (P-DeSIP) Removing potential proxy bias and (U-DeSIP) Removing unresolved bias from Ding et al. (2022). Both are restricted to debiasing based on gendered terms.

**WEAT\*: Information Retained**    This is also an intrinsic information preservation metric that was proposed by Dev et al. (2021a). WEAT\* is an extension of WEAT (Caliskan et al., 2017) to measure gendered information (male vs. female associations) retained after debiasing word embedding directly. Here the two target set of gendered words ($X$ : {man, male, boy, brother, him, his, son} and $Y$ : {woman, female, girl, sister, her, hers, daughter}) are kept constant.

The main modification to WEAT is that instead of the attribute set of words ($A$ and $B$) being stereotypical (e.g., $A$: male-biased occupations and $B$ female-biased ones), $A$ and $B$ are definitionally gendered ($A$ male and $B$ female) so we want the score $s(X, Y, A, B)$ (see Appendix E) to be large. Following, Dev et al. (2021a) we use $A$, $B$ as he-she on WEAT\*(1), as definitionally gendered words (e.g., father, actor, and mother, actress) on WEAT\*(2) and as gendered names (e.g., james, ryan and emma, sophia) on WEAT\*(3). A higher score indicates more meaningful gendered (male vs. female) information is preserved.

All four debiasing techniques (LP, HD, INLP, OSCaR) retain the least gendered information. Thus they destroy the meaningful gendered (male vs. female) information in the word embedding. However, ISR outperforms all four debiasing techniques on WEAT\* and even improves the Baseline (Glove without debiasing). ISR, therefore inherently retains the key information while improving bias removal.

Table 14: Results on WEAT\*, a metric to measure how much correctly gendered information is retained after debiasing an embedding. \*: represents results reported from original OSCaR paper Dev et al. (2021a). A higher score indicates more meaningful gendered (male vs. female) information is preserved.

| Pretrained Model | Method | WEAT\*(1) | WEAT\*(2) | WEAT\*(3) |
|---|---|---|---|---|
| Glove | Baseline\* | 1.845 | 1.856 | 1.874 |
| | LP\* | 0.385 | 1.207 | 1.389 |
| | HD\* | 1.667 | 1.554 | 0.822 |
| | INLP\* | 0.789 | 1.368 | 0.873 |
| | OSCaR\* | 1.361 | 1.396 | 1.543 |
| | ISR | 1.885 | 1.918 | 1.927 |

**Bias-by-projection Task.**    We compute the dot product between gender direction, $\overrightarrow{he} - \overrightarrow{she}$ and top 50,000 most frequent words. The resulting absolute dot products scores are then averaged to get the Bias-by-projection score. After debiasing the word embedding, if the Bias-by-projection score is closer to 0, then we have effectively removed all evidence of gender bias. Hard debiasing achieved the lowest Bias-by-projection score of 0.0002, which is not surprising since it projects the word embedding away from the gender direction and takes a more aggressive approach to remove all gender information from the embedding representation. See Table 15 Note that ISR has nearly the largest score on this task. We actually do not view this as a negative since it means it is able to retain meaningful associations with concepts (e.g., grandma with she, and grandpa with he) that may be useful for natural language understanding, such as document summarization, question answering, and information extraction or co-reference resolution.

**Sembias Analogy Task.**    This aims at finding the word pair that is the best analogy to the pair $(he, she)$ by considering these four options: a gender-specific word pair, e.g., *(waiter,waitress)*; a gender-stereotype word pair, e.g., *(doctor,nurse)*; and two highly-similar, bias-free word pairs, e.g. *(dog, cat)* Zhao et al. (2018). The dataset used for the Sembias Analogy Task contains 440

instances. 40 instances denoted by SemBias* or are not used during training. Other than the P-DeSIP and U-DeSIP designed for this task Ding et al. (2022), ISR achieves the highest score accuracy in identifying gender-specific word pairs. See Table 15

Table 15: Average the absolute projection bias of the top 50,000 most frequent words

|         | Bias-by-proj | SemBias | SemBias* |
|---------|--------------|---------|----------|
| Orig.   | 0.0435       | 0.7955  | 0.7750   |
| LP      | 0.0463       | 0.8273  | 0.7500   |
| HD      | 0.0002       | 0.5045  | 0.5250   |
| INLP    | 0.0961       | 0.2432  | 0.1500   |
| P-DeSIP | 0.0038       | 0.9523  | 0.9750   |
| U-DeSIP | 0.0038       | 0.9090  | 0.5000   |
| OSCaR   | 0.0384       | 0.8182  | 0.9750   |
| SR      | 0.0441       | 0.8591  | 0.8500   |
| iOSCaR  | 0.0232       | 0.2795  | 0.0000   |
| ISR     | 0.0729       | 0.8750  | 0.9250   |

**Word Similarity Tasks.** In as much as we are interested in removing or eliminating bias or stereotypical association from word embeddings, we want to ensure the semantic information within the word embeddings are preserved. The word similarity task was conducted using the following English word similarity benchmarks: RG65 Rubenstein & Goodenough (1965), WordSim-353 Finkelstein et al. (2001), Rarewords Luong et al. (2013), MEN Bruni et al. (2012), MTurk-287 Radinsky et al. (2011), and MTurk-771 Halawi et al. (2012), SimLex-999 Hill et al. (2015), and SimVerb-3500 Gerz et al. (2016). We measure the semantic information preserved on the glove embedding and all the debiased GloVe models. The Spearman rank coefficient scores shows that ISR and iOSCaR retains useful structures and semantic information from the original embeddings. See Table 16

Table 16: Word Similarity Score (Spearman rank coefficient)

|         | RG65   | WS     | RW     | MEN    | MT-287 | MT-771 | SimLex | SimVerb |
|---------|--------|--------|--------|--------|--------|--------|--------|---------|
| Orig.   | 0.7656 | 0.6014 | 0.4113 | 0.7373 | 0.6333 | 0.6499 | 0.3708 | 0.2305  |
| LP      | 0.7521 | 0.6068 | 0.4217 | 0.7427 | 0.6375 | 0.6534 | 0.3793 | 0.2373  |
| HD      | 0.7492 | 0.6093 | 0.4247 | 0.7445 | 0.6387 | 0.6516 | 0.3919 | 0.2404  |
| INLP    | 0.6991 | 0.6086 | 0.4551 | 0.7533 | 0.6369 | 0.6505 | 0.4165 | 0.2780  |
| P-DeSIP | 0.7794 | 0.6856 | 0.3970 | 0.7484 | 0.6452 | 0.6741 | 0.3765 | 0.2286  |
| U-DeSIP | 0.7734 | 0.6828 | 0.3956 | 0.7478 | 0.6455 | 0.6731 | 0.3756 | 0.2273  |
| OSCaR   | 0.7618 | 0.6015 | 0.4116 | 0.7360 | 0.6346 | 0.6492 | 0.3698 | 0.2309  |
| SR      | 0.7614 | 0.6024 | 0.4127 | 0.7360 | 0.6354 | 0.6500 | 0.3691 | 0.2303  |
| iOSCaR  | 0.7614 | 0.5988 | 0.4116 | 0.7335 | 0.6336 | 0.6459 | 0.3700 | 0.2311  |
| ISR     | 0.7479 | 0.6009 | 0.4136 | 0.7304 | 0.6412 | 0.6490 | 0.3645 | 0.2285  |

## D  ADDITIONAL EXPERIMENT WITH THREE CONCEPTS

We also perform an experiment on rectifying three subspaces with ISR using different concepts. In this case we consider definitionally gendered male/female terms (GT: $A, B$), pleasant/unpleasant terms (P/U: $X, Y$), and statistically gendered male/female names (GN: $R, S$). We use our large word lists of these terms. This is potentially interesting again because someone's name may not align with the statistically most likely gender association, and it may have an unwanted, unpleasant connotation. So one may want to perform rectification with both gender association and an unpleasant association, another intersectional issue. This experiment is also interesting because gendered terms and statistically gendered names generate subspaces with a large dot product (initially larger than 0.8). As we observe, ISR faces a greater challenge in both reducing this association, and also retaining the information, because of the overlapping space they occupy but still obtain near-orthogonal

subspaces. We observe that $v_1$ (gendered terms) and $v_2$ (pleasant/unpleasant) have the smallest dot product, so rectify these first within the iteration.

Table 17: WEAT Scores and dot products after Debiasing

| Iteration | WEAT | | | dot product | | |
|---|---|---|---|---|---|---|
| | GT vs GN | GT vs P/U | GN vs P/U | GT vs GN | GT vs P/U | GN vs P/U |
| Orig. | 1.6706 | 0.3337 | 1.1118 | 0.8237 | 0.0729 | 0.1245 |
| Iter 1 | 1.1999 | 0.0944 | 0.5396 | 0.5478 | 0.0194 | 0.0533 |
| Iter 2 | 0.6376 | 0.1738 | 0.3225 | 0.3213 | 0.0421 | 0.0327 |
| Iter 3 | 0.3140 | 0.1574 | 0.2174 | 0.1753 | 0.0397 | 0.0225 |
| Iter 4 | 0.1554 | 0.1223 | 0.1505 | 0.0913 | 0.0305 | 0.0154 |
| Iter 5 | 0.0788 | 0.0897 | 0.1034 | 0.0464 | 0.0210 | 0.0101 |
| Iter 6 | 0.0414 | 0.0644 | 0.0714 | 0.0233 | 0.0135 | 0.0064 |
| Iter 7 | 0.0230 | 0.0467 | 0.0504 | 0.0116 | 0.0082 | 0.0039 |
| Iter 8 | 0.0140 | 0.0354 | 0.0372 | 0.0058 | 0.0047 | 0.0022 |
| Iter 9 | 0.0095 | 0.0285 | 0.0291 | 0.0029 | 0.0026 | 0.0013 |
| Iter 10 | 0.0073 | 0.0246 | 0.0245 | 0.0014 | 0.0014 | 0.0007 |

Table 17 shows the WEAT score of the ISR mechanism during the iterations 1 through 10, measured on the full set. We observe that all pairwise WEAT scores decrease significantly (to about 0.02) after 10 iterations. Similarly, Table 17 also shows the pairwise dot products throughout 10 iterations. We observe that the largest initial dot-product pair (Gendered Terms vs. Gendered Names) starts very similar at 0.8237 and decreases to 0.0014 after 10 iterations, similar to the values achieved for other pairs. Finally, in Table 18, we show the SWEAT scores for each concept pair throughout the process. This shows the information retained as a function of the Self-WEAT scores. We observe that while Pleasant/Unpleasant retains most of its SWEAT score, we do see a noticeable decrease for gendered terms and statistically gendered names. This is likely since they start with a dot-product of 0.82, they are very correlated, and some words overlap along the defined subspace. So some non-trivial warping is necessary to orthogonalize the concepts.

Table 18: SWEAT Scores after Debiasing

| | Before | Iter 1 | Iter 2 | Iter 3 | Iter 4 | Iter 5 | Iter 6 | Iter 7 | Iter 8 | Iter 9 | Iter 10 |
|---|---|---|---|---|---|---|---|---|---|---|---|
| GT | 1.7674 | 1.6793 | 1.4597 | 1.3844 | 1.3492 | 1.2856 | 1.2078 | 1.3435 | 1.2313 | 1.2101 | 1.1030 |
| GN | 1.9041 | 1.843 | 1.7859 | 1.7165 | 1.6545 | 1.6639 | 1.6172 | 1.6013 | 1.5955 | 1.5903 | 1.5917 |
| Pleasant/Un | 1.8762 | 1.8767 | 1.8636 | 1.8718 | 1.8762 | 1.8575 | 1.8616 | 1.8797 | 1.8771 | 1.8744 | 1.8647 |

### D.1 10 ITERATIONS ON NN-GT-P/U EXPERIMENT.

We also show the results after all 10 iterations for the example with Gendered Terms(M/F) (GT), Nationality associated Names (USA/Mexico) (NN), and Pleasant Unpleasant terms (P/U). The WEAT scores and dot products are in Table 19, and the SWEAT scores are in Table 20.

## E   DEFINITION OF WEAT

The Word Embedding Association Test (WEAT) Caliskan et al. (2017) is the default measurement of association among paired concepts from word lists, via their embedding. It takes two target word lists $X$ and $Y$ (e.g., male and female terms) and two attribute words lists $A$ and $B$ (e.g., pleasant and unpleasant words). For each word $w \in X \cup Y$ we compute the association of $w$ with set $A, B$ as:

$$s\left(w, A, B\right) = \frac{1}{|A|} \sum_{a \in A} \cos\left(a, w\right)$$

$$- \frac{1}{|B|} \sum_{b \in B} \cos\left(b, w\right)$$

Table 19: WEAT Scores and dot products after Debiasing

| | WEAT | | | dot product | | |
|---|---|---|---|---|---|---|
| Iteration | GT vs NN | GT vs P/U | NN vs P/U | GT vs NN | GT vs P/U | NN vs P/U |
| Orig. | 0.1797 | 0.3337 | 1.1506 | 0.0589 | 0.0729 | 0.1721 |
| Iter 1 | 0.1157 | 0.129 | 0.6195 | 0.0395 | 0.0273 | 0.0598 |
| Iter 2 | 0.0657 | 0.0442 | 0.3146 | 0.0252 | 0.0104 | 0.0204 |
| Iter 3 | 0.0316 | 0.0113 | 0.1974 | 0.0157 | 0.0041 | 0.0070 |
| Iter 4 | 0.0097 | 0.0015 | 0.1564 | 0.0096 | 0.0017 | 0.0024 |
| Iter 5 | 0.0040 | 0.0067 | 0.1423 | 0.0058 | 0.0007 | 0.0008 |
| Iter 6 | 0.0124 | 0.0089 | 0.1375 | 0.0035 | 0.0003 | 0.0003 |
| Iter 7 | 0.0175 | 0.0099 | 0.1359 | 0.0021 | 0.0001 | 0.0001 |
| Iter 8 | 0.0205 | 0.0103 | 0.1353 | 0.0012 | 0.0001 | 0.0000 |
| Iter 9 | 0.0223 | 0.0105 | 0.1351 | 0.0007 | 0.0000 | 0.0000 |
| Iter 10 | 0.0234 | 0.0106 | 0.1351 | 0.0004 | 0.0000 | 0.0000 |

Table 20: SWEAT Scores after Debiasing

| | Orig. | Iter 1 | Iter 2 | Iter 3 | Iter 4 | Iter 5 | Iter 6 | Iter 7 | Iter 8 | Iter 9 | Iter 10 |
|---|---|---|---|---|---|---|---|---|---|---|---|
| GT | 1.7674 | 1.7692 | 1.7502 | 1.7637 | 1.7745 | 1.7545 | 1.7427 | 1.7445 | 1.7433 | 1.7484 | 1.7471 |
| NN | 1.7289 | 1.7298 | 1.7459 | 1.7592 | 1.7711 | 1.7386 | 1.7317 | 1.7352 | 1.7569 | 1.7141 | 1.7716 |
| Plea/Un | 1.8762 | 1.8768 | 1.8648 | 1.8715 | 1.8761 | 1.8603 | 1.8619 | 1.8803 | 1.8780 | 1.8745 | 1.8662 |

Then we average this value $s(w, A, B)$ over all words in the $X$ and $Y$ set to get:

$$s\left(X, Y, A, B\right) = \frac{1}{|X|} \sum_{x \in X} s\left(x, A, B\right)$$
$$- \frac{1}{|Y|} \sum_{y \in Y} s\left(y, A, B\right)$$

$s(X, Y, A, B)$ is then normalized by the standard deviation of $s(w, A, B), \forall w \in X \cup Y$ to get the WEAT score. The WEAT score typical lies in the range $[-1, 1]$ and a value closer to 0 indicates less biased association.

## F  WORD LISTS USED IN EXPERIMENTS

For many experiments we begin with small bespoke word lists used to define concepts. Extended from Caliskan et al. (2017).

- **Definitionally Gendered Terms [Gen(M/F)]:**
  - **Male Terms**: male, man, boy, brother, he, him, his, son
  - **Female Terms**: female, woman, girl, sister, she, her, hers, daughter
- **Pleasant and Unpleasant Words [Please/Un]:**
  - **Pleasant**: caress, freedom, health, love, peace, cheer, friend, heaven, loyal, pleasure, diamond, gentle, honest, lucky, rainbow, diploma, gift, honor, miracle, sunrise, family, happy, laughter, paradise, vacation
  - **Unpleasant**: abuse, crash, filth, murder, sickness, accident, death, grief, poison, stink, assault, disaster, hatred, pollute, tragedy, bomb, divorce, jail, poverty, ugly, cancer, evil, kill, rotten, vomit
- **Career and Family Words [Career/Family]:**
  - **Career**: executive, management, professional, corporation, salary, office, business, career

- **Family**: home, parents, children, family, cousins, marriage, wedding, relatives
- **Math, Science, and Arts Words [Sci/Art],[Math/Art]**:
    - **Math**: math, algebra, geometry, calculus, equations, computation, numbers, addition
    - **Science**: science, technology, physics, chemistry, einstein, nasa, experiment, astronomy
    - **Arts**: poetry, art, dance, literature, novel, symphony, drama, sculpture
- **Common, Statistical Gendered Names [Name(M/F)]**
    - **Male Names**: John, Paul, Mike, Kevin, Steve, Greg, Jeff, Bill
    - **Female Names**: Amy, Joan, Lisa, Sarah, Diana, Kate, Ann, Donna
- **Names associated with Cultural origins [Name(E/A)]**
    - **European American names**: brad, brendan, geoffrey, greg, brett, jay, matthew, neil, todd, allison, anne, carrie, emily, jill, laurie, kristen, meredith, sarah
    - **African American names**: darnell, hakim, jermaine, kareem, jamal, leroy, rasheed, tremayne, tyrone, aisha, ebony, keisha, kenya, latoya, tamika, tanisha
- **Flower and Insects [Flower/Insect]**
    - **Flowers**: aster, clover, hyacinth, marigold, poppy, azalea, crocus, iris, orchid, rose, daffodil, lilac, pansy, tulip, buttercup, daisy, lily, peony, violet, carnation, magnolia, petunia, zinnia
    - **Insect**: ant, caterpillar, flea, locust, spider, bedbug, centipede, fly, maggot, tarantula, bee, cockroach, gnat, mosquito, termite, beetle, cricket, hornet, moth, wasp, dragonfly, roach, weevil
- **Musical Instrument and Weapons [Music/Weap]**:
    - **Musical Instruments**: bagpipe, cello, guitar, lute, trombone, banjo, clarinet, harmonica, mandolin, trumpet, bassoon, drum, harp, oboe, tuba, bell, fiddle, harpsichord, piano, viola, bongo, flute, horn, saxophone, violin
    - **Weapons**: arrow, club, gun, missile, spear, axe, dagger, harpoon, pistol, sword, blade, dynamite, hatchet, rifle, tank, bomb, firearm, knife, shotgun, teargas, cannon, grenade, mace, slingshot, whip

## F.1    LARGER WORD LISTS FROM LIWC

For performing test/train splits, it is often necessary to arrange for larger word lists, so each half of a split has sufficient words to define a concept. For this, we identify some very large word lists (with sometimes hundreds of words) from LIWC Pennebaker et al. (2001). These word lists initially contained many words with wild card symbols (*) to represent many ways a word can end (e.g., -er, -ed, -est, -es). For each such word, we select all possible matching words in the larger word lists of the embedding.

However, these sets are still quite noisy, and some of the words are tangentially related to the concept or very rare, so the embedding representative is not reliable. Including them ultimately does not improve the estimation of the concept in the word embedding. We found it was better to select a careful and central subset of these larger word lists. To do this, we start with the mean of our associated smaller word list (from those above, a touchstone word when the bespoke word list is unavailable) and select the 100 closest words to that mean (including ones in the smaller bespoke list). The result word lists are presented next:

- Definitionally Gendered Terms [Gen(M/F)]:
    - **Male (100 Words)**: father, son, brother, man, his, him, he, boy, himself, husband, uncle, grandfather, nephew, grandson, sons, guy, men, dad, boys, male, sir, king, brothers, boyfriend, prince, stepfather, fellow, guys, businessman, gentleman, earl, mr, grandparents, brother-in-law, duke, paternal, son-in-law, father-in-law, monk, fathers, knight, buddy, daddy, stepson, nephews, congressman, uncles, bull, fathered, husbands, chairman, fiance, masculine, patriarch, colt, salesman, godfather, cowboy, grandsons, bachelor, macho, spokesman, schoolboy, kings, males, gentlemen,

boyhood, monastery, statesman, grandpa, lad, countrymen, papa, boyish, fraternity, princes, cowboys, penis, dude, baritone, monks, knighted, knights, lions, bulls, prostate, businessmen, strongman, mister, czar, roh, deer, manly, gonzales, dukes, stud, manhood, brethren, paternity, colts

- **Female (100 Words)**: woman, her, mother, girl, she, daughter, wife, sister, herself, grandmother, girlfriend, daughters, aunt, mom, female, niece, lady, girls, women, actress, hers, sisters, granddaughter, boyfriend, princess, mistress, queen, heroine, bride, mothers, maid, waitress, jane, housewife, wives, nun, actresses, feminine, fiancee, ladies, stepmother, stepdaughter, diva, fiance, lesbian, goddess, feminist, duchess, countess, husbands, mrs, maternal, madame, womb, mama, schoolgirl, madam, grandma, sister-in-law, businesswoman, hostess, socialite, heiress, maiden, ballerina, witch, daughter-in-law, mommy, mum, godmother, congresswoman, motherhood, spokeswoman, moms, aunts, queens, nieces, tomboy, feminism, females, uterus, granddaughters, matron, boyfriends, maternity, femininity, heroines, divorcee, princesses, mimi, sorority, landlady, dame, matriarch, dowry, chairwoman, lesbians, girlish, grandmothers, vagina

- Pleasant and Unpleasant Words [Please/Un]:

  - **Pleasant (100 Words)**: good, pretty, kind, honest, well, beautiful, surprisingly, generous, nice, certainly, wonderful, better, decent, handsome, sure, strong, happy, easy, rich, truly, lovely, excellent, like, charming, intelligent, loving, warm, thoughtful, gentle, polite, fun, perfect, enjoy, smart, healthy, funny, proud, thanks, interesting, great, giving, bright, best, love, wonderfully, definitely, confident, amazingly, terrific, comfortable, passionate, energetic, true, cool, liked, helpful, brilliant, perfectly, lively, importantly, fine, elegant, talented, fair, important, appreciate, exciting, enthusiastic, clever, cheerful, welcome, promising, opportunity, respect, respectful, wise, pleasant, hope, promise, gracious, entertaining, likes, brave, wealthy, enjoyed, sincere, enjoying, impressed, pleased, impressive, surely, gorgeous, impression, sweet, pleasing, useful, eager, promises, caring, loved

  - **Unpleasant (100 Words)**: bad, stupid, ugly, weak, worse, poor, cruel, arrogant, awful, nasty, terribly, unfair, rude, pathetic, lousy, ineffective, foolish, ignorant, dangerous, miserable, wrong, terrible, disgusting, unfortunately, unfortunate, difficult, unattractive, horrible, abusive, cynical, incompetent, timid, greedy, shockingly, unpleasant, annoying, lazy, inadequate, disappointing, selfish, frustrating, vicious, depressing, brutal, dumb, scared, scary, ridiculous, shameful, pitiful, sad, aggressive, outrageous, desperate, boring, sorry, afraid, harsh, vulnerable, crazy, immoral, worried, confusing, obnoxious, problematic, unhappy, grossly, complain, dreadful, embarrassing, frightening, insecure, hurt, useless, uncomfortable, awkward, confused, dangerously, painful, appalling, careless, discouraging, risky, hurting, heartless, frustrated, deceptive, ineffectual, demeaning, horribly, angry, sick, depressed, messy, worrying, wicked, ridiculously, unacceptable, suffer, irritating

- Career and Family Words [Career/Family]:

  - **Career (100 Words)**: lawyer, director, professor, scientist, economist, executive, consultant, assistant, politician, businessman, banker, worked, retired, colleague, associate, administrator, hired, adviser, governor, researcher, entrepreneur, chairman, president, mathematician, senior, writer, scholar, wrote, office, advisor, psychologist, investigator, ceo, institute, counsel, biologist, diplomat, secretary, working, department, manager, editor, lawrence, law, university, teacher, lawyers, doctor, interview, analyst, managing, producer, lecturer, research, succeeded, company, finance, lawmaker, industrialist, consulting, dean, congressman, studied, staff, leader, graduate, publisher, economics, legal, colleagues, associates, financier, worker, administration, political, job, written, developer, government, employee, librarian, committee, work, boss, succeed, graduated, reporter, agency, trader, works, business, client, directors, programmer, student, bank, supervisor, leading, mentor, agent

  - **Family (100 Words)**: parents, family, mother, daughters, relatives, daughter, wife, grandparents, families, husband, marriage, wedding, siblings, grandmother, father, married, mothers, marry, wives, sister, sons, son, divorced, aunt, husbands, grandchildren, cousins, cousin, baby, mom, pregnant, sisters, spouses, brother, niece, spouse,

divorce, marriages, fathers, babies, dad, granddaughter, uncle, grandfather, brothers, widowed, widow, honeymoon, aunts, maternal, fiancee, weddings, parent, fiance, maternity, stepfather, nephews, uncles, nephew, paternal, nieces, grandchild, pregnancy, grandson, parental, stepmother, moms, widows, sibling, folks, grandmothers, grand-daughters, divorcing, dads, paternity, parenting, in-laws, grandma, nanny, widower, marries, stepdaughter, motherhood, stepchildren, fathered, grandsons, sister-in-law, pregnancies, divorces, grandparent, kin, nannies, daddy, daughter-in-law, grandkids, mama, brother-in-law, mommy, mum, parenthood

- Statistically Gendered Names [Name(M/F)]:
  - **Male (100 Words)**: kevin, john, paul, scott, chris, brian, ryan, anderson, michael, wilson, terry, walker, larry, keith, davis, gary, james, joe, eric, allen, david, jason, bennett, sean, bruce, graham, thomas, peter, russell, jack, stephen, bryan, tony, robert, richard, steven, jerry, frank, patrick, martin, mark, ian, anthony, andy, clark, simon, jon, adam, taylor, jay, sullivan, andrew, brett, jonathan, lewis, reid, quinn, danny, parker, alan, matthew, dennis, mitchell, justin, jimmy, eddie, ellis, randy, riley, charlie, dean, shane, johnny, derek, elliott, george, neil, bradley, jeremy, francis, curtis, casey, nelson, trevor, hayes, harrison, alex, aaron, kyle, jackson, darren, roy, jamie, hunter, fisher, roger, lawrence, blake, william, marshall
  - **Female (100 Words)**: sarah, lisa, amy, kate, jennifer, linda, laura, mary, elizabeth, anne, jane, katherine, julie, maggie, helen, rebecca, jessica, emily, lauren, margaret, lucy, caroline, rachel, michelle, emma, katie, diana, marie, louise, barbara, anna, martha, catherine, ellen, melissa, alice, kathleen, sara, claire, christine, julia, patricia, stephanie, leslie, karen, cynthia, frances, hannah, natalie, dorothy, vanessa, amanda, jacqueline, nancy, elaine, samantha, sophie, annie, judith, nicole, kelly, christina, megan, joanna, ashley, naomi, molly, irene, maria, melanie, ruth, brenda, sylvia, carolyn, parker, holly, eliza, nina, deborah, gwen, marilyn, sandra, esther, veronica, fiona, edith, eleanor, alicia, erin, eileen, evelyn, alison, princess, kathryn, bridget, claudia, chloe, eva, angela, abigail

- Achieve/Anxious Words [Achieve/Anx]:
  - **Achieve (100 Words)**: achieve, able, ability, effort, accomplish, successful, success, gain, efforts, better, enable, goal, obtain, failed, opportunity, achieved, plan, fulfill, achieving, improve, try, accomplished, trying, work, strategy, determined, leadership, enabling, goals, unable, progress, ambitious, advantage, win, enabled, skills, create, determination, fail, purpose, best, plans, skill, first, leaders, working, attain, confident, failing, capability, overcome, successfully, failure, challenge, abilities, advance, succeed, capabilities, promote, fails, winning, potential, initiative, tried, opportunities, lead, lose, victory, obtaining, ambition, earn, planning, ahead, achievements, capable, initiatives, challenging, confidence, team, resolve, creating, fulfilling, improving, leader, gaining, ambitions, fulfilled, achievement, competitive, solve, challenges, leading, limited, solution, efficient, acquire, successes, planned, strive, promotion
  - **Anxious (100 Words)**: fear, nervous, worried, fearful, afraid, scared, anxious, frightened, panic, anxiety, worry, confused, uneasy, upset, uncomfortable, insecure, terrified, alarmed, panicked, apprehensive, fears, fearing, worrying, nervousness, hesitant, embarrassed, distress, disturbed, upsetting, unsure, panicky, alarm, distraught, vulnerable, discomfort, dread, panicking, frightening, worries, desperate, unease, uneasiness, uncertainty, reluctant, shaken, paranoia, impatient, avoid, overwhelmed, ashamed, paranoid, doubt, insecurity, irritated, scare, tension, feared, risk, threatening, scary, uncertain, tense, desperation, phobia, obsessed, shaking, apprehension, unsettling, turmoil, awkward, startled, stress, unsettled, irrational, distressed, desperately, confusing, risks, embarrassment, shame, vulnerability, suspicious, neurotic, timid, restless, aversion, terrifying, irritable, threat, irritation, risked, scares, threats, frighten, alarming, disturbing, irritating, obsessive, horrible, alarms

- Statistically American/Mexican Names :
  - **American Names (100 Words) [Name(M/F)]**: david, michael, john, chris, alex, daniel, james, mike, robert, kevin, mark, brian, anthony, jason, joe, eric, andrew, ryan, paul, richard, william, victor, jonathan, matt, joseph, tony, steve, justin, brandon, jeff, matthew, scott, nick, christopher, steven, andrea, josh, jay, sam, adam,

thomas, jim, joshua, tim, tom, frank, george, aaron, dan, martin, mary, jennifer, jessica, michelle, lisa, sarah, ana, elizabeth, laura, ashley, linda, karen, stephanie, sandra, melissa, amanda, nancy, patricia, emily, nicole, amy, carmen, susan, rosa, angela, diana, rachel, martha, kelly, anna, brenda, sara, julie, kim, barbara, katie, monica, claudia, lauren, gloria, veronica, kathy, heather, samantha, teresa, cindy, kimberly, sharon, christina, vanessa

– **Mexican Names (100 Words) [Name(M/F)]**: jose, juan, luis, carlos, jesus, jorge, alejandro, miguel, angel, manuel, eduardo, fernando, francisco, antonio, javier, ricardo, oscar, pedro, roberto, alberto, mario, sergio, gerardo, arturo, cesar, armando, omar, diego, alfredo, edgar, raul, enrique, hector, ivan, rafael, julio, gabriel, adrian, pablo, gustavo, andres, josé, jaime, marco, hugo, guillermo, alexis, alan, erick, cristian, maria, guadalupe, lupita, alejandra, karla, adriana, isabel, fernanda, silvia, gabriela, mariana, mari, daniela, erika, paola, margarita, karina, alicia, alma, norma, leticia, angelica, blanca, rosario, rocio, gaby, carolina, dulce, lorena, valeria, cristina, ale, miriam, yolanda, mayra, araceli, marisol, esmeralda, irma, luz, paty, sofia, elena, rosy, maribel, cecilia, alondra, juana, tere, liliana

We note that the OSCaR paper (Dev et al., 2021a) also makes an effort to keep the test and training words disjoint. However, they take a slightly different approach in using a similarly large evaluation set, but using a bespoke (carefully chosen by hand) training set. For instance, for the Male/Female gender direction, they primarily use just the words "he" and "she." And for occupations they define a subspace using the top principal component of a small word list (scientist, doctor, nurse, secretary, maid, dancer, cleaner, advocate, player, banker). Their splits are not random.

## G DOT PRODUCT SCORES

We also show that the dot product score converges to 0 for ISR but not for iOSCaR on the other concept pairs we experimented with. These results appear in Tables 21, 22, 23, 24, 25, and 26.

Table 21: Dot Products Before and After Debiasing on Large Lists and Test/Train Split

|  | Gen(M/F) & Please/Un | | Gen(M/F) & Career/Family | | Name(M/F) & Please/Un | |
| Iteration | iOSCaR | ISR | iOSCaR | ISR | iOSCaR | ISR |
| --- | --- | --- | --- | --- | --- | --- |
| Before | 0.0683 | 0.0683 | 0.3276 | 0.3276 | 0.1231 | 0.1231 |
| Iter 1 | 0.1392 | 0.0214 | 0.1299 | 0.1736 | 0.0385 | 0.0466 |
| Iter 2 | 0.3300 | 0.0067 | 0.0536 | 0.0911 | 0.0679 | 0.0177 |
| Iter 3 | 0.5327 | 0.0021 | 0.1374 | 0.0479 | 0.1462 | 0.0067 |
| Iter 4 | 0.6388 | 0.0007 | 0.2590 | 0.0253 | 0.2643 | 0.0026 |
| Iter 5 | 0.6274 | 0.0002 | 0.4106 | 0.0134 | 0.4082 | 0.0010 |
| Iter 6 | 0.6381 | 0.0001 | 0.5447 | 0.0071 | 0.5158 | 0.0004 |
| Iter 7 | 0.5830 | 0.0000 | 0.6329 | 0.0038 | 0.5419 | 0.0001 |
| Iter 8 | 0.6036 | 0.0000 | 0.6437 | 0.0020 | 0.7061 | 0.0001 |
| Iter 9 | 0.5807 | 0.0000 | 0.6294 | 0.0011 | 0.6508 | 0.0000 |
| Iter 10 | 0.6224 | 0.0000 | 0.6708 | 0.0006 | 0.6693 | 0.0000 |

Table 22: Dot Products Before and After Debiasing on Large Lists and Test/Train Split

| | Name(M/F) & Career/Family | | Gen(M/F) & Name(M/F) | | Gen(M/F) & Achieve/Anx | |
| Iteration | iOSCaR | ISR | iOSCaR | ISR | iOSCaR | ISR |
|---|---|---|---|---|---|---|
| Before | 0.4525 | 0.4525 | 0.7650 | 0.7650 | 0.1599 | 0.1599 |
| Iter 1 | 0.0840 | 0.1571 | 0.5361 | 0.4475 | 0.0426 | 0.0401 |
| Iter 2 | 0.2442 | 0.0533 | 0.1781 | 0.2344 | 0.1574 | 0.0101 |
| Iter 3 | 0.4225 | 0.0182 | 0.2002 | 0.1176 | 0.2949 | 0.0026 |
| Iter 4 | 0.6117 | 0.0062 | 0.5466 | 0.0578 | 0.4624 | 0.0007 |
| Iter 5 | 0.6443 | 0.0021 | 0.6645 | 0.0282 | 0.6298 | 0.0002 |
| Iter 6 | 0.6598 | 0.0007 | 0.4505 | 0.0137 | 0.6080 | 0.0000 |
| Iter 7 | 0.5916 | 0.0003 | 0.6122 | 0.0067 | 0.6778 | 0.0000 |
| Iter 8 | 0.6187 | 0.0001 | 0.4379 | 0.0032 | 0.6248 | 0.0000 |
| Iter 9 | 0.5845 | 0.0000 | 0.6000 | 0.0016 | 0.6938 | 0.0000 |
| Iter 10 | 0.6249 | 0.0000 | 0.4402 | 0.0008 | 0.6170 | 0.0000 |

Table 23: Dot Products Before and After Debiasing on Large Lists and Test/Train Split

| | Name(M/F) & Career/Family | | Career/Family & Achieve/Anx | |
| Iteration | iOSCaR | ISR | iOSCaR | ISR |
|---|---|---|---|---|
| Before | 0.1528 | 0.1528 | 0.3080 | 0.3080 |
| Iter 1 | 0.0756 | 0.0529 | 0.1587 | 0.0860 |
| Iter 2 | 0.2432 | 0.0182 | 0.3573 | 0.0240 |
| Iter 3 | 0.4388 | 0.0063 | 0.5393 | 0.0067 |
| Iter 4 | 0.6358 | 0.0022 | 0.6625 | 0.0019 |
| Iter 5 | 0.6273 | 0.0008 | 0.6001 | 0.0005 |
| Iter 6 | 0.6515 | 0.0003 | 0.6188 | 0.0002 |
| Iter 7 | 0.5878 | 0.0001 | 0.5600 | 0.0000 |
| Iter 8 | 0.6516 | 0.0000 | 0.6004 | 0.0000 |
| Iter 9 | 0.5308 | 0.0000 | 0.6008 | 0.0000 |
| Iter 10 | 0.6473 | 0.0000 | 0.5871 | 0.0000 |

Table 24: Dot Products Before and After Debiasing on Large Lists and No Test/Train Split

| | Gen(M/F) & Please/Un | | Gen(M/F) & Career/Family | | Name(M/F) & Please/Un | |
| Iteration | iOSCaR | ISR | iOSCaR | ISR | iOSCaR | ISR |
|---|---|---|---|---|---|---|
| Before | 0.0729 | 0.0729 | 0.3724 | 0.3724 | 0.1245 | 0.1245 |
| Iter 1 | 0.1345 | 0.0245 | 0.1623 | 0.2054 | 0.0390 | 0.0506 |
| Iter 2 | 0.3041 | 0.0082 | 0.0272 | 0.1108 | 0.0116 | 0.0206 |
| Iter 3 | 0.5067 | 0.0028 | 0.0978 | 0.0597 | 0.0666 | 0.0083 |
| Iter 4 | 0.7002 | 0.0009 | 0.2299 | 0.0322 | 0.1500 | 0.0034 |
| Iter 5 | 0.5623 | 0.0003 | 0.4155 | 0.0174 | 0.2865 | 0.0014 |
| Iter 6 | 0.7593 | 0.0001 | 0.6236 | 0.0094 | 0.4804 | 0.0006 |
| Iter 7 | 0.5065 | 0.0000 | 0.7827 | 0.0051 | 0.6782 | 0.0002 |
| Iter 8 | 0.7257 | 0.0000 | 0.5824 | 0.0027 | 0.8155 | 0.0001 |
| Iter 9 | 0.4361 | 0.0000 | 0.7558 | 0.0015 | 0.6497 | 0.0000 |
| Iter 10 | 0.6803 | 0.0000 | 0.5310 | 0.0008 | 0.7983 | 0.0000 |

Table 25: Dot Products Before and After Debiasing on Large Lists and No Test/Train Split

| Iteration | Name(M/F) & Career/Family | | Gen(M/F) & Name(M/F) | | Gen(M/F) & Achieve/Anx | |
|---|---|---|---|---|---|---|
| | iOSCaR | ISR | iOSCaR | ISR | iOSCaR | ISR |
| Before | 0.4609 | 0.4609 | 0.8237 | 0.8237 | 0.1899 | 0.1899 |
| Iter 1 | 0.0613 | 0.1678 | 0.6974 | 0.5592 | 0.0342 | 0.0497 |
| Iter 2 | 0.2215 | 0.0593 | 0.4636 | 0.3324 | 0.1270 | 0.0130 |
| Iter 3 | 0.4052 | 0.0211 | 0.0823 | 0.1809 | 0.2418 | 0.0034 |
| Iter 4 | 0.6148 | 0.0075 | 0.3933 | 0.0939 | 0.4175 | 0.0009 |
| Iter 5 | 0.7672 | 0.0027 | 0.6993 | 0.0478 | 0.6212 | 0.0002 |
| Iter 6 | 0.5479 | 0.0010 | 0.3143 | 0.0241 | 0.7801 | 0.0001 |
| Iter 7 | 0.7046 | 0.0003 | 0.6668 | 0.0121 | 0.5792 | 0.0000 |
| Iter 8 | 0.5372 | 0.0001 | 0.2871 | 0.0061 | 0.7521 | 0.0000 |
| Iter 9 | 0.7171 | 0.0000 | 0.6770 | 0.0030 | 0.5265 | 0.0000 |
| Iter 10 | 0.4506 | 0.0000 | 0.2343 | 0.0015 | 0.7142 | 0.0000 |

Table 26: Dot Products Before and After Debiasing on Large Lists and No Test/Train Split

| Iteration | Career/Family & Please/Un | | Career/Family & Achieve/Anx ) | |
|---|---|---|---|---|
| | iOSCaR | ISR | iOSCaR | ISR |
| Before | 0.1449 | 0.1449 | 0.3266 | 0.3266 |
| Iter 1 | 0.0594 | 0.0510 | 0.152 | 0.0963 |
| Iter 2 | 0.2533 | 0.0179 | 0.3464 | 0.0283 |
| Iter 3 | 0.4763 | 0.0063 | 0.5327 | 0.0083 |
| Iter 4 | 0.6847 | 0.0022 | 0.6983 | 0.0025 |
| Iter 5 | 0.6141 | 0.0008 | 0.5338 | 0.0007 |
| Iter 6 | 0.7299 | 0.0003 | 0.7458 | 0.0002 |
| Iter 7 | 0.4133 | 0.0001 | 0.4704 | 0.0001 |
| Iter 8 | 0.5718 | 0.0000 | 0.7129 | 0.0000 |
| Iter 9 | 0.5828 | 0.0000 | 0.4141 | 0.0000 |
| Iter 10 | 0.5625 | 0.0000 | 0.6971 | 0.0000 |

## H  STANDARD DEVIATION SCORES

Table 27: Sample Standard Deviation Score on Large Lists and Test/Train Split.

| Concept1 | Concept2 | Orig. | LP | HD | INLP | OSCaR | SR | iOSCaR | ISR |
|---|---|---|---|---|---|---|---|---|---|
| Gen(M/F) | Please/Un | 0.1215 | 0.2434 | 0.1916 | 0.1928 | 0.1629 | 0.1911 | 0.1435 | 0.2259 |
| Gen(M/F) | Career/Family | 0.0926 | 0.3712 | 0.1998 | 0.2449 | 0.2096 | 0.1655 | 0.2443 | 0.2472 |
| Name(M/F) | Please/Un | 0.1513 | 0.3207 | 0.3480 | 0.3191 | 0.2456 | 0.3403 | 0.2966 | 0.4630 |
| Name(M/F) | Career/Family | 0.0376 | 0.2493 | 0.2370 | 0.2066 | 0.2071 | 0.1900 | 0.2124 | 0.3389 |
| Gen(M/F) | Name(M/F) | 0.0245 | 0.2676 | 0.2121 | 0.1616 | 0.1716 | 0.0360 | 0.1513 | 0.0614 |
| Gen(M/F) | Achieve/Anx | 0.0738 | 0.3594 | 0.2221 | 0.1865 | 0.2071 | 0.1500 | 0.2124 | 0.1875 |
| Career/Family | Please/Un | 0.0719 | 0.3964 | 0.4006 | 0.3573 | 0.3866 | 0.3394 | 0.4133 | 0.5318 |
| Career/Family | Achieve/Anx | 0.1343 | 0.2979 | 0.3094 | 0.3277 | 0.1800 | 0.2700 | 0.1754 | 0.3337 |

## I  WORD LIST SIZE ABLATION STUDIES

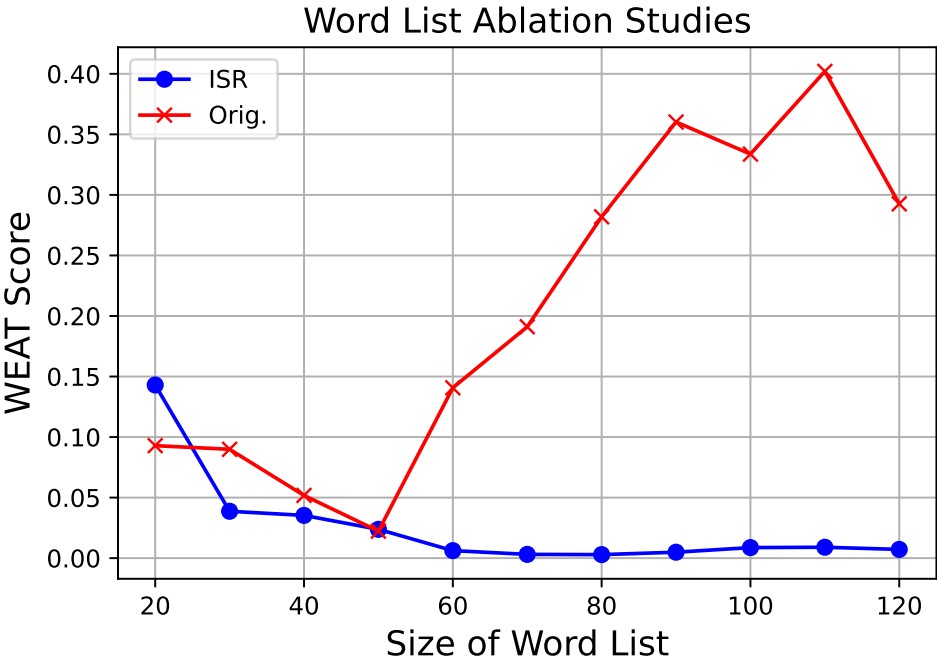

Figure 1: We perform an ablation study to understand how sensitive ISR is to the size of the list of words used during training.

It covers the WEAT score on the standard Gendered Terms vs. Pleasant/Unpleasant terms large word lists at various sizes. As for how we selected the top 120 words, we ordered all words from the larger list using their distance from the words in the associated small list from the original IAT that WEAT is based on. We consider subsets of words from size 20 to 120. We observe that the WEAT score after applying ISR decreases as the word list increases in size (from about 0.14 at $k = 20$ to about 0.01 at $k = 60$) and is fairly stable up to $k = 120$. The minimum occurs around $k = 80$. In contrast, the WEAT score before applying ISR mostly increases until about $k = 90$. So our choice of $k = 100$ provides about the best discriminatory power but is not very sensitive in the range of $k = 80$ to $k = 120$.

