# OpenReview forum: "Interpretable Debiasing of Vectorized Language Representations with Iterative Orthogonalization"
_ICLR.cc/2023/Conference — ICLR 2023 poster_

### Official Review · Reviewer_Zber · 2022-10-21

**Confidence:** 4
**Clarity, Quality, Novelty And Reproducibility:** Novelty
**Correctness:** 3
**Technical Novelty And Significance:** 2
**Empirical Novelty And Significance:** 2
**Recommendation:** 6

**Strength And Weaknesses:**

------ Strengths ------

The proposed modifications seem to improve both concept disentangling and preserving information about concepts.
Multiple subspace debiasing is an interesting direction of research.

------ Weaknesses ------

Choice of evaluation methods is questionable, presentation is highly misleading and overselling the approach.

Presentation:

1) starting from the abstract, contribution list, etc, the paper states that it’s them who propose using disentangling of concepts rather than removing information from the embeddings. However, (1) this was first done by OSCaR they modify, (2) quality of preserving information is not noticeably better than that of OSCaR.

2) In the contribution list, they write “we can even perform a train-test split experiment (which is rarely performed in this domain)...” - again, the train-test split has already been done when evaluating the original OSCaR.

All in all, it gives the impression that the authors are trying to take credit for some of the OSCaR’s core ideas.

Evaluation:

1) Preserving concept information has been first done and evaluated by OSCaR. However, the authors highlight preserving as a novel thing and propose their own evaluation set. Why not follow the original evaluation by OSCaR?

2) Evaluation is only intrinsic while other works (e.g., OSCaR) have also used extrinsic evaluation.


**Summary Of The Paper:**

The paper proposes a mechanism of post-hoc modification of word embeddings such that problematic associations between concepts (i.e., biases) are disentanged while information about concepts is still retained. The proposed approach is a modification of previously proposed OSCaR that rectifies to desired directions (male-female vs occupations) so that they become orthogonal and rotates the rest of the components. Modifications made in the current work are: (1) choosing the central point of the rotation, (2) rerunning OSCaR iteratively to obtain better orthogonality, (3) extension to multiple concept debiasing.

**Summary Of The Review:**

Overall, the paper leaves mixed feelings: while proposed modifications do seem to improve disentanglement quality, the paper seems to try taking credit for previous work.

---

> ### Author Response · Authors · 2022-11-15
> **Response to Reviewer Zber**
>
> We thank the reviewers for their insightful comments. Sorry for the slow reply, we ran a significant number of new experiments, based on your comments, which took time to complete. We provide our responses below.
>
> 1. *starting from the abstract, contribution list, etc, the paper states that it’s them who propose using disentangling of concepts rather than removing information from the embeddings. However, (1) this was first done by OSCaR they modify, (2) quality of preserving information is not noticeably better than that of OSCaR.*
>
> While this is subtle, we feel our writing is not wrong or dishonest. OSCaR does not augment the data so if the concepts are relearned the subspaces are orthogonal, whereas we demonstrate that ISR does. We feel this is an essential contribution, especially for extending the ideas to multiple concepts and for improved interpretability.
>
> Moreover, before we list our contributions in the introduction, we introduce OSCaR and note that it initiated this idea and that our approach extends it in several ways. The second bullet directly contrasts with OSCaR in how we observe that its output data does not represent orthogonal subspaces. The fourth contribution contrasts with other more common approaches (HD and INLP) and highlights the new SWEAT evaluation method. We feel we give OSCaR fair credit for its important contributions. If there is a different way of describing this that you suggest, please let us know.
>
>
> Regarding (2) the quality of preserving information, there are two ways to interpret this.  If the information of concern is generic -- that is not specific to the debiasing augmentation -- then basically all methods in this class see almost no change (perhaps with the exception of INLP).  This is addressed elsewhere in responses to reviewer comments.  However, on tasks specific to concepts being augmented, then we disagree with the statement "quality of preserving information is not noticeably better than that of OSCaR."  For instance, in our SWEAT test designed to measure this (Table 7), ISR's association values match the original data up to about 0.01.  In contrast, the scores for OSCaR drop consistently and an average of 0.11.  Moreover, if you iterate OSCaR without centering (iOSCaR), then the scores drop significantly by an average of 1.08, demonstrating that the centering step we add is essential towards preserving the information (that is, the error in one step is non-trivial and can amplify).
>
>
> 2. *In the contribution list, they write “we can even perform a train-test split experiment (which is rarely performed in this domain)...” - again, the train-test split has already been done when evaluating the original OSCaR.*
>
> OSCaR does not quite do this.  It is true that the OSCaR paper does have a section on "Keeping train/test separate."  But this is not a test-train split in the traditional sense, and definitely not a random one.  This section in the OSCaR paper comments on how they used just "he/she" as the training set or a small set of works for occupations (scientist, doctor, nurse, secretary, maid, dancer, cleaner, advocate, player, banker).  This set was carefully chosen (from personal communication with the OSCaR authors) to provide a representative subspace.  It was not randomly selected from a large enough representative list as we have done.
>
> We agree the distinction with OSCaR could be made more clear, so we have updated the paper in the following ways:
> - in the contribution list, it now states that "we perform a *randomized* train-test split. "
>  - In Section 3.2 on Evaluation using a Test/Train Split, we add, "E.g., many papers (Bolukbasi etal 2016, Dev etal 2021) select only the he-she pair to train Gen(M/F). "
>  - In the Appendix F1, where the larger word lists are discussed (and space is not so constrained), we added the following paragraph: "We note that the OSCaR paper (Dev et al. 2021) also makes an effort to keep the test and training words disjoint.  However, they take a slightly different approach using a similarly large evaluation set but a bespoke (carefully chosen by hand) training set.  For instance, for the Male/Female gender direction, they primarily use just the words "he" and "she."  And for occupations, they define a subspace using the top

---

> > ### Comment · Reviewer_Zber · 2022-11-25
> > **Comments after rebuttal**
> >
> > I thank the authors for further working on the paper and improving presentation and mentions of previous work. I remain mostly positive about the paper and keeping my overall recommendation at "6: marginally above the acceptance threshold".

---

### Official Review · Reviewer_sTGU · 2022-10-25

**Confidence:** 2
**Clarity, Quality, Novelty And Reproducibility:** The paper is well written. The method…
**Correctness:** 3
**Technical Novelty And Significance:** 3
**Empirical Novelty And Significance:** 3
**Recommendation:** 6

**Strength And Weaknesses:**

- Overall, it is a well written paper. General audience who don't directly work on language embedding can understand this paper. The research topic is timely and the proposed method looks promising.
- Numerical experiments show significant improvement over existing methods.
- About information preservation of ISR: suppose concept A has been orthogonalized by ISR w.r.t. concept B, and concept C is orthogonalized w.r.t. concept B with ISR, would the information between A and C be preserved? One example may be that gender is orthogonalized w.r.t. un/pleasant, and race (e.g. white and black) is orthogonalized with un/pleasant, would the information between gender and race be preserved in terms of, e.g. WEAT metric?

**Summary Of The Paper:**

This paper proposes an iterative recentering-rotation method to orthogonalize word embeddings to remove bias. Merits of the new method are demonstrated on debias embedding of words in standard dataset as well as large language models. Furthermore, it is shown that the information is preserved.

**Summary Of The Review:**

The contribution is timely and the merits are corroborated by strong empirical results. I have only a minor concern on information preservation, which could be important for some applications.

---

> ### Author Response · Authors · 2022-11-15
> **Response to Reviewer sTGU**
>
> We thank the reviewers for their insightful comments. Sorry for the slow reply, we ran a significant number of new experiments, based on your comments, which took time to complete. We provide our responses below.
>
> *About information preservation of ISR: suppose concept A has been orthogonalized by ISR w.r.t. concept B, and concept C is orthogonalized w.r.t. concept B with ISR, would the information between A and C be preserved? One example may be that gender is orthogonalized w.r.t. un/pleasant, and race (e.g. white and black) is orthogonalized with un/pleasant, would the information between gender and race be preserved in terms of, e.g. WEAT metric?*
>
> We informally experimented with this approach with the goal of making all three concepts orthogonal.  While our initial goal was the dot-products approaching 0, we informally observed this WEAT performance as well.  In this sense, it performed less well and less predictably than the method proposed in the paper, where concept C is orthogonalized with respect to the subspace defined by concepts A and B.  This ensures all three concepts are orthogonal.
>
> However, it seems you are asking for something different.  If the correlation between concepts A and C is retained if concepts pairs A & B and B & C are orthogonalized.  We found that it was not necessarily the case on simple trials, but we have not performed a careful and large study to empirically demonstrate this yet.

---

### Official Review · Reviewer_r1GB · 2022-10-25

**Confidence:** 2
**Correctness:** 3
**Technical Novelty And Significance:** 2
**Empirical Novelty And Significance:** 3
**Recommendation:** 6

**Clarity, Quality, Novelty And Reproducibility:**

Clarity: This work's clarity is good. It is helpful when things are emphasized for clarity with italics (eg "The underlying operation is based on a rotation, and our insight is <i>how to choose the central point that the data is rotated around.<i>"

Quality: The quality of the work is fine. There are some methodological concerns (eg evaluate utility) and quibbles (eg report std dev for experiments run many times), but there are no obvious errors in the experiments.

Novelty: The novelty of this work is somewhat low. It extends an existing algorithm and evaluates it on an existing task/dataset.

Reproducibility: The code for this work is not available.

**Strength And Weaknesses:**

STRENGTHS
- Unlike most other debiasing attempts, this work measures performance with a train/test split.
- The ISR model (this work) scores much better on the debiasing task than previous works do (95% reduction in bias) on the WEAT task.
- Iteratively running OSCaR is a good baseline to compare against, because it allows for disentangling where the "secret sauce" of ISR is or isn't coming from.

WEAKNESSES
- Nearly all of this work is dedicated to measuring if the bias is removed. Very little measures if the vectors are still useful.
- The only experiment to measure utility is Table 7's measurement of "Information Preserved." However, this doesn't measure something like downstream prediction performance or ability to reconstruct the original vectors, but instead measures a statistic of the original vectors and of the new vectors, and compares how much has changed. However, it seems like it could be possible to produce new vectors that are very different from the old ones but whose SWEAT scores happen to be similar. That would be reported as information-preserving.
- Instead of spending real state extending the approach (eg to sentence vectors, to three concepts, etc), it would be more useful to spend more time demonstrating the new vectors maintain their utility despite the orthogonalization of certain subspaces.

MINOR-ISH
- This paper uses a novel approach for defining subspaces based on the mean of a small set of coded poles (eg for male-female, words like him/he/king vs her/she/queen). This seems as sensible as any other approach, but the authors assert this method is more "reliable" than other approaches (eg normal direction of a linear classifier or the first principal component of the union of a pair) without defining what they mean by "reliable."
- On page 5, although an experiment is run 10 times, and its average is reported... the standard deviation is not also reported. If space is a concern, it could go in the appendix.
- Although this work claims that it can be extended to debias multiple subspaces which could "potentially address[] intersectional issues," this misunderstands the standard concept of intersectionality. Intersectionality is about when the sum is greater than its parts (a model that hires white women and black men but not black women might not be discriminating w.r.t. race  or w.r.t. gender but w.r.t race+gender). However, this method attempts to make each subspace orthogonal to each other, which is (arguably) useful in some settings but is explicitly counter to the concept of effects from intersectionality.


**Summary Of The Paper:**

This paper presents an approach for "debiasing" word vectors (and sentence vectors). Because it is rotation-based, this method does not destroy information the way projection-based methods do. The main approach is to extend the OSCaR subspace correction by applying it iteratively.

**Summary Of The Review:**

I have revised my assessment from marginal-reject to marginal-accept because of the additional experiments and clarifications from the authors.

---

> ### Author Response · Authors · 2022-11-15
> **Response to Reviewer r1GB**
>
> 3. *Instead of spending real state extending the approach (eg to sentence vectors, to three concepts, etc), it would be more useful to spend more time demonstrating the new vectors maintain their utility despite the orthogonalization of certain subspaces.*
>
> We feel that, in principle, this class of approaches does not change much performance on generic tasks and is fairly well-known.  However, the downstream tasks of sentence vectors on contextual embeddings were new and exciting.  Moreover, it required new innovation to apply to multiple subspaces, which is also an important and novel step.  It certainly took us a while to figure out the right way to approach this -- and it was not addressed in the OSCaR paper.  Moreover, we feel the ability to do this, which relies on better orthogonalization than OSCaR can achieve, is very useful for issues that involve interpretability or which concepts to orthogonalize (the answer can now be "many").
>
> 4. *This paper uses a novel approach for defining subspaces based on the mean of a small set of coded poles (eg for male-female, words like him/he/king vs her/she/queen). This seems as sensible as any other approach, but the authors assert this method is more "reliable" than other approaches (eg normal direction of a linear classifier or the first principal component of the union of a pair) without defining what they mean by "reliable."*
>
> While the paper shows experiments on standardized and often larger word lists and experiments, we tried many small variations on smaller word lists in designing our approaches.  This exploratory approach did not have a great experimental design -- not something we would feel great reporting formally.  We then fixed a design and chose our proposed approach to experiment against existing methods.  While we kept some variants (like iOSCaR and SR), we did not show all, including how we chose the subspaces.
>
> But we feel most other approaches are indeed not as stable.  For instance, a linear classifier of an SVM in high dimension will depend on a large number of support vectors, and any one outlier point can rotate the normal vector significantly.  And the first principal component might not be aligned from one set to the other if the groups have high variance.
>
> 5. *On page 5, although an experiment is run 10 times, and its average is reported... the standard deviation is not also reported. If space is a concern, it could go in the appendix.*
>
> We have added this result in Appendix H.
>
> 6. *Although this work claims that it can be extended to debias multiple subspaces which could "potentially address intersectional issues," this misunderstands the standard concept of intersectionality. Intersectionality is about when the sum is greater than its parts (a model that hires white women and black men but not black women might not be discriminating w.r.t. race or w.r.t. gender but w.r.t race+gender). However, this method attempts to make each subspace orthogonal to each other, which is (arguably) useful in some settings but is explicitly counter to the concept of effects from intersectionality.*
>
> We do feel that this approach may be helpful in understanding and examining intersectionality. For instance, does orthogonalizing concepts in such representations remove intersectional effects, or do they persist? (We hypothesize they will persist.)  If we first orthogonalize, can we then decompose an intersectional effect along the two (or more) components to measure the influence of each? (maybe).
>
> As you note, we only say "potentially address" these issues.  We believe the above questions may be an interesting direction, but we have not yet come up with an experimental evaluation of this we are satisfied with.  We leave this for future work, building on the ideas we present here.
>
>
> 7. *Novelty: The novelty of this work is somewhat low. It extends an existing algorithm and evaluates it on an existing task/dataset.*
>
> We admit the changes to the OSCaR method are fairly small.  but we believe the choices are subtle, and the improvement is essential.  In particular, the ability to achieve very-nearly orthogonal concepts is essential towards interpretability and multi-concept debiasing.
>
> The paper also adds new classes of experiments.
>
> 8. *Reproducibility: The code for this work is not available.*
>
> Note that we had stated we will publically release code after the anonymous phase.  In the meantime we have uploaded a preliminary anonymized version.
>
> 9. *One shortcoming of this work is that there is insufficient work done to demonstrate the new vectors maintain their utility after being "debiased", which could be done a few ways such as clinical downstream performance or demonstrating the old vectors could be recovered from the new ones.*
>
> We hope the explanations above and the new experiment have addressed this concern.

---

> ### Author Response · Authors · 2022-11-15
> **Response to Reviewer r1GB**
>
> We thank the reviewers for their insightful comments. Sorry for the slow reply, we ran a significant number of new experiments, based on your comments, which took time to complete. We provide our responses below.
>
> 1. *Nearly all of this work is dedicated to measuring if the bias is removed. Very little measures if the vectors are still useful.*
>
> In addition to Section 3.4 (with more results in Table 10, Appendix), which measures information preservation on lists specific to the topic being augmented, we also include a generic part-of-speech (POS) tagging experiment in the Appendix (was Table 11, now Table 13). This shows the change in F1, Precision, and Recall on these tasks for LP, HD, INLP, and ISR. The largest change is 0.0127 and is typically about 0.001. This is typical in these debiasing techniques, which augment representations in a low dimensional subspace, leaving most other concepts unchanged and leading to minor changes to performance on generic tasks. Hence, we felt it was better to focus on other evaluations in the main body of the paper.
>
> Moreover, we have added another set of experiments from the OSCaR paper.  These consider a Natural Language Inference (NLI) task using RoBERTa as the LLM before (Baseline) and after LP, HD, INLP, and OSCaR are performed.  We added ISR to these results.  First, on the generic results (Table 11: Dev F1, Test F1 columns), we see that before augmenting the methods, perform 91.2 & 90.5 on Dev & Test rests.  All previous methods are with 0.3 of both values.  ISR achieves 91.5 and 90.8, which is the best in both cases, but we view it as not a notable improvement.  The main takeaway is that on generic tasks, the performance is rarely changed in a meaningful way.
>
> This experiment also adds a targeted task (like the SEAT evaluation we develop) that measures on sentences and tasks specific to the type of data and concepts being augmented. On these tasks measuring bias attenuation (Table 11), ISR basically matches OSCaR, which showed a large improvement over all other methods. Table 12 shows information preserved on targetted tasks. Here most methods show very little degradation (with the possible exception of INLP, which removes about 35 dimensions, e.g., dropping from a score of 94.9 to 92.8 on Net Entailment). Notably, ISR universally achieves the largest score in each measurement, including attaining a 97.4 on Net Entailment and 100.0 on Fractional Entailment.
>
>
> 2. *The only experiment to measure utility is Table 7's measurement of "Information Preserved." However, this doesn't measure something like downstream prediction performance or ability to reconstruct the original vectors, but instead measures a statistic of the original vectors and of the new vectors, and compares how much has changed. However, it seems like it could be possible to produce new vectors that are very different from the old ones but whose SWEAT scores happen to be similar. That would be reported as information-preserving.*
>
> As described in response to the point above, the original paper showed performance on a generic POS tagging task in Appendix C. We have added an experiment on an NLI task also in Appendix C.
>
> The explanation as to why OSCaR (or most of these methods with the possible exception of INLP) does not dramatically change the representations, and hence their performance on downstream tasks, is as follows.  These representations are high-dimension (i.e., 300 for GloVe and 1024 for one layer of RoBERTa), but the methods only augment the representations along a 1- or 2-d subspaces.  If we change the basis, this is the equivalent of only changing 1 or 2 coordinates.  From this guiding perspective, we believe it is intuitive that only minor changes in generic downstream performance are observed.

---

### Official Review · Reviewer_PKsy · 2022-11-01

**Confidence:** 2
**Correctness:** 3
**Technical Novelty And Significance:** 1
**Empirical Novelty And Significance:** 2
**Recommendation:** 6

**Clarity, Quality, Novelty And Reproducibility:**

The paper is difficult to read and needs significant revision. There is not much novelty in the proposed solution.

**Strength And Weaknesses:**

Strengths:
      1. Simple approach which seems to do well in the evaluation.

Weaknesses:
       1. The proposed improvement is targeted for a specific debiasing technique and is not relevant for others.
       2. No mathematical proof for the convergence of iterative method is provided.


Questions:
        1. How robust are WEAT/SEAT scores wrt the size of the lists?
	2. Table 4 --> INLP is consistently better than ISR and iOSCAR. Why?
	3. Why is there significant difference in the absolute effect size for BERT and RoBERTa?
	4. How sensitive is the proposed method to the list of words (both number and specific words) used for training? Have you done an ablation study on the no of words used?
	5. Why's the idea of vectorize sentences containing those words contained in a Wikipedia dump and average of sentences from Wikipedia containing the concept words correct representation of the concept? Might have concepts unrelated to the male vs. female gender and these might get lost.
	6. In Table 7, SWEAT score increased for ISR relative to Orig (1.8705 vs 1.8677) for Achieve/Anx Gen(M/F). Why?
        7. How's SWEAT score a measure of information preserved?
        8. Can you prove that the debiasing technique proposed in the submission doesn't introduce new bias inadvertently?

**Summary Of The Paper:**

This work addresses the problem of removing bias from word embeddings where the challenge is to retain information while removing bias associated with a concept. Specifically it looks addressing the shortcomings of a previous work, OSCaR, which looks at the linear subspace formed by two concepts (e.g. gender and occupation) and applies transformation to the subspace in an attempt to make the two concepts orthogonal. However, the resulting vectorial representation of the two concepts is not guaranteed to be orthogonal. The current work attempts to make the representations orthogonal by iteratively applying OSCaR with a centering step. Experimental results are provided to demonstrate the superiority of the proposed improvement over OSCaR and other debiasing techniques. The key claim is that unlike previous techniques based on projections, there is no loss of information  with the proposed method.

The modification to OSCaR proposed in the current work consist primarily of finding a point other than origin for rotation step. This center point is chosen to be the midpoint of midpoints of concept pairs. However, if  the centering, projection, graded rotation, un-project, and un-center operations are applied only once, the resulting subspaces are not always fully orthogonal. So the work proposed to repeat these steps iteratively till orthogonality is achieved.



**Summary Of The Review:**

The work is a minor improvement of a previous work.

---

> ### Author Response · Authors · 2022-11-15
> **Response to Reviewer PKsy**
>
> 5. *Why is there significant difference in the absolute effect size for BERT and RoBERTa?*
>
> Note that, in Table 6, the BERT average starts at 0.620, and ISR reduces it to 0.190 (a difference of 0.430).  RoBERTa average starts at 0.940, and ISR reduces it to 0.385 (a difference of 0.555).  We think these differences are fairly similar.
>
> 6. *How sensitive is the proposed method to the list of words (both number and specific words) used for training? Have you done an ablation study on the no of words used?*
>
> We have added a new plot in Appendix I to show the sensitivity to the number of words used (discussed above).  We have also added (Table 27 in Appendix H) the standard deviation associated with Table 4 where we do random test/train splits.  This shows something about the stability of the results with respect to splits.  This confirms it is not a very stable experiment (because of the non-iid nature of words in natural language), although we feel the average is representative and useful.
>
> 7. *Why's the idea of vectorize sentences containing those words contained in a Wikipedia dump and average of sentences from Wikipedia containing the concept words correct representation of the concept? Might have concepts unrelated to the male vs. female gender and these might get lost.*
>
> We choose this method and experimental setup from previous papers (May et al. 2019, Meade et al. 2022).  This indeed could be an issue; we feel that as an average of taken over many sentences, it provides a useful signal.
>
> Note we have also provided other ways to evaluate these methods. This now includes an experiment using an existing Natural Language Inference task (Appendix C.1 & C.2) that designs template sentences to measure which controls this more carefully. The trade-off between these two types of evaluation is the SentenceDebias task uses "natural sentences," whereas the NLI task uses constructed sentences that may be less representative of real language. Whichever you prefer, we now include experiments of both types.
>
> 8. *In Table 7, SWEAT score increased for ISR relative to Orig (1.8705 vs 1.8677) for Achieve/Anx Gen(M/F). Why?*
>
> These measurements are over natural language representations and several randomized trials. We expect some small variability in answers and feel this difference is within that variability. We consider these changes in the third or fourth digit of precision to be not substantial.
>
> 9. *How's SWEAT score a measure of information preserved?*
>
> Just as the WEAT score measures the effect size under a permutation test on the splits, a similar perspective can be used on the SWEAT test.  In this case, we *desire* a large effect size, so lists A1 and A2 are more associated with each other than with lists B1 and B2.
>
> 10. *Can you prove that the debiasing technique proposed in the submission doesn't introduce new bias inadvertently?*
>
> In this paper, we take the perspective that the representation model is constructed with a black box.  This makes the approach quite general but means we do not assume much of anything about the relative representations of each word -- other than perhaps certain conceptual groups of works cluster in some way.
>
> This perspective makes it hard to prove much at all about side effects that may happen.  However, our reasoning about why this is not likely to occur is as follows.  These representations are high-dimensional, and as has been observed over many studies, concepts tend to align as cluster groups or linear subspaces.  These ultimately mean these concepts are low-dimensional and, in high-dimensional likely do not interfere with each other.  The method we develop (ISR) and most others in this general family of techniques (e.g., HD, LP, OSCaR) operate on a 1 or 2-dimensional subspace and leave the remaining aspects of the representation unchanged.  Hence, if other concepts are primarily represented in the null space of these concepts (over 99% of the space), then they are likely to be mostly or completely unchanged.
>
> We, in fact, observe this in the mostly unchanged performance on the POS Tagging and NLI tasks in the Appendix.
>
> 11. *The paper is difficult to read and needs significant revision. There is not much novelty in the proposed solution.  The work is a minor improvement of a previous work.*
>
> We disagree with this statement.  While this work builds on existing techniques (don't almost all papers), and the changes are subtle, they are carefully designed.   Moreover, we feel the resulting improvements are very important.  In particular, the fact that ISR converges to having orthogonal concepts is essential in uses in multi-concept debiasing and interpretation.

---

> ### Author Response · Authors · 2022-11-15
> **Response to Reviewer PKsy**
>
> We thank the reviewers for their insightful comments. Sorry for the slow reply, we ran a significant number of new experiments, based on your comments, which took time to complete. We provide our responses below.
>
> 1. *The proposed improvement is targeted for a specific debiasing technique and is not relevant for others.*
>
> We compare against quite a few existing debiasing techniques:
>  - HD (Bolukbasi etal 2016)
>  - LP (Dev etal 2019)
>  - INLP (Ravfogel etal 2020)
>  - SentenceDebias (Liang etal 2020)
>  - OSCaR (Dev etal 2021)
>  - CDA (Zmigord etal 2019)
>  - DROPOUT (Webster etal 2020)
>
> These are applied to contextual and non-contextual language representational models, including the auto-regressive model (GPT-2) in the appendix.
>
> Moreover, we believe this is broadly useful for debiasing, regularizing, and interpreting learned vectorized representations -- where evaluation mechanisms are less developed and tangible.
>
> 2. *No mathematical proof for the convergence of iterative method is provided.*
>
> This is a fair critique.  It is a question we have approached but do not have an answer to.  We have witnessed non-monotonic behavior, but we believe it may be possible to prove something about the limiting behavior.  A formal proof seems challenging, but we believe we have provided sufficient empirical evidence of the effectiveness of OSCaR.
>
> We point out that among other techniques in this family, there is little in the way of formal analysis.
>
> 3. *How robust are WEAT/SEAT scores wrt the size of the lists?*
>
> We have added a small ablation study in Appendix H.
> It covers the WEAT score on the standard Gendered Terms vs. Pleasant/Unpleasant terms large word lists at various sizes.  As for how we selected the top 100 words, we ordered all words from the larger list using their distance from the words in the associated small list from the original IAT that WEAT is based on.  We consider subsets of words from size 20 to 120.  We observe that the WEAT score after applying ISR decreases as the word list increases in size (from about 0.14 at k=20 to about 0.01 at k=60) and is fairly stable up to k=120.  The minimum occurs around k=80.  In contrast, the WEAT score before applying ISR mostly increases until about k=90.   So our choice of k=100 provides about the best discriminatory power but is not very sensitive in the range of k=80 to k=120.
>
>
> 4. *Table 4 --> INLP is consistently better than ISR and iOSCAR. Why?*
>
> Yes, INLP performs better on 6 of 8 experiments in this one experiment.  As we note in the paper, this is likely because it destroys over 30 dimensions worth of information, and in this noisier setting, that appears to carry over to held-out data better than ISR.  We note that this results in INLP having more loss of information than ISR, as demonstrated in the SEAT tests to evaluate information retained.
>
> If one were to run INLP indefinitely until there are 0 dimensions remaining, clearly, that would reduce all of these values to exactly 0.  But it would also retain zero information in the representation and not be a good option.  The paper makes the case that ISR is the superior overall approach.

---

### Decision · Program_Chairs · 2023-01-20

**Decision:**

Accept: poster

**Justification For Why Not Higher Score:**

n/a

**Justification For Why Not Lower Score:**

Clearly there is an agreement the paper is reasonably well developed.

**Metareview: Summary, Strengths And Weaknesses:**

This paper presents a way to debias the embedding space of language representations. The reviewers had some concerns about the paper, but the authors properly addressed them in their comments. The reviewers are now, overall, quite happy to see this paper at ICLR.

**Note From Pc:**

if the above contains the word "oral" or "spotlight" please see: "oral" presentation means -> notable-top-5% and "spotlight" means -> notable-top-25%. As stated in our emails, we are disassociating presentation type from AC recommendations

**Summary Of Ac-Reviewer Meeting:**

there was no need for a meeting, as things were mostly resolved on the message board.